# Effects of prescribed CMIP6 ozone on simulating the Southern Hemisphere atmospheric circulation response to ozone depletion

Ioana Ivanciu[1], Katja Matthes[1,2], Sebastian Wahl[1], Jan Harlaß[1], and Arne Biastoch[1,2]

[1]GEOMAR Helmholtz Centre for Ocean Research Kiel, Kiel, Germany
[2]Faculty of Mathematics and Natural Sciences, Christian-Albrechts Universität zu Kiel, Kiel, Germany

**Correspondence:** Ioana Ivanciu (iivanciu@geomar.de)

**Abstract.** The Antarctic ozone hole has led to substantial changes in the Southern Hemisphere atmospheric circulation, such as the strengthening and poleward shift of the mid-latitude westerly jet. Ozone recovery during the twenty-first century is expected to continue to affect the jet's strength and position, leading to changes in the opposite direction compared to the twentieth century and competing with the effect of increasing greenhouse gases. Simulations of the Earth's past and future climate, such as those performed for the Coupled Model Intercomparison Project Phase 6 (CMIP6), require an accurate representation of these ozone effects. Climate models that use prescribed ozone fields lack the important feedbacks between ozone chemistry, radiative heating, dynamics, as well as transport. In addition, when the prescribed ozone field was not generated by the same model to which it is prescribed, the imposed ozone hole is inconsistent with the simulated dynamics. These limitations ultimately affect the climate response to ozone depletion. This study investigates the impact of prescribing the ozone field recommended for CMIP6 on the simulated effects of ozone depletion in the Southern Hemisphere. We employ a new, state-of the-art coupled climate model, FOCI, to compare simulations in which the CMIP6 ozone is prescribed with simulations in which the ozone chemistry is calculated interactively. At the same time, we compare the roles played by ozone depletion and by increasing concentrations of greenhouse gases in driving changes in the Southern Hemisphere atmospheric circulation, using a series of historical sensitivity simulations. FOCI captures the known effects of ozone depletion, simulating an austral spring and summer intensification of the mid-latitude westerly winds and of the Brewer-Dobson circulation in the Southern Hemisphere. Ozone depletion is the primary driver of these historical circulation changes in FOCI. The austral spring cooling of the polar cap in the lower stratosphere in response to ozone depletion is weaker in the simulations that prescribe the CMIP6 ozone field. We attribute this weaker response to a prescribed ozone hole which is different to the model dynamics and is not collocated with the simulated polar vortex, altering the strength and position of the planetary wavenumber one. As a result, the dynamical contribution to the ozone-induced austral spring lower stratospheric cooling is suppressed, leading to a weaker cooling trend. Consequently, the intensification of the polar night jet is also weaker in the simulations with prescribed CMIP6 ozone. In contrast, the differences in the tropospheric westerly jet response to ozone depletion fall within the internal variability present in the model. The persistence of the Southern Annular Mode is shorter in the prescribed ozone chemistry simulations. The results obtained with the FOCI model suggest that climate models which prescribe the CMIP6 ozone field still simulate a weaker Southern Hemisphere stratospheric response to ozone depletion compared to models that calculate the ozone chemistry interactively.

# 1 Introduction

Anthropogenic emissions of ozone depleting substances (ODS), in particular Chlorofluorocarbons (CFCs), led to a steep decline in stratospheric ozone concentrations since the 1980s. The strongest ozone depletion occured in austral spring above Antarctica. There, the particularly low temperatures inside the winter polar vortex enable the formation of polar stratospheric clouds (PSC). Upon the arrival of sunlight in spring, heterogeneous chlorine photochemistry on the surface of PSC makes chlorine particularly effective at destroying ozone (e.g., Solomon, 1999). As a result, the ozone hole develops every spring in the Antarctic stratosphere, with profound impacts for the Southern Hemisphere (SH) climate. Observations (e.g., Randel and Wu, 1999; Thompson and Solomon, 2002; Randel et al., 2009; Young et al., 2013) and model simulations (Mahlman et al., 1994; Arblaster and Meehl, 2006; Gillett and Thompson, 2003; Stolarski et al., 2010; Perlwitz et al., 2008; Son et al., 2010; McLandress et al., 2010; Polvani et al., 2011; Young et al., 2013; Eyring et al., 2013; Keeble et al., 2014) consistently show a cooling of the Antarctic lower stratosphere in austral spring and summer during the last decades of the twentieth century due to decreased radiative heating as a result of ozone depletion. This cooling led to important changes in the dynamics of the SH. Lower polar cap temperatures resulted in an increased meridional temperature gradient between the cold polar cap and the relatively warmer mid-latitudes. Consequently, the spring stratospheric polar vortex strengthened (Thompson and Solomon, 2002; Gillett and Thompson, 2003; Arblaster and Meehl, 2006; McLandress et al., 2010; Thompson et al., 2011; Keeble et al., 2014) and its breakdown was delayed by about two weeks (Waugh et al., 1999; Langematz et al., 2003; McLandress et al., 2010; Previdi and Polvani, 2014; Keeble et al., 2014). This enabled an intensification of the planetary wave activity propagating into the stratosphere, resulting in an enhancement of the Brewer-Dobson circulation (BDC) in austral summer (Li et al., 2008, 2010; Oberländer-Hayn et al., 2015; Polvani et al., 2018; Abalos et al., 2019).

At the same time, the strengthening of the stratospheric westerlies extended downward affecting the tropospheric jet, which intensified with a lag of one to two months, in austral summer (Thompson and Solomon, 2002; Gillett and Thompson, 2003; Perlwitz et al., 2008; Son et al., 2010; Eyring et al., 2013). The intensification of the stratospheric and tropospheric jets was accompanied by a concurrent positive trend in the Southern Annular Mode (SAM, Thompson and Solomon, 2002; Gillett and Thompson, 2003; Marshall, 2003; Perlwitz et al., 2008; Fogt et al., 2009; Thompson et al., 2011). The surface westerlies strengthened on their poleward side and weakened on their equatorward side, therefore shifting towards higher latitudes during the austral summer (Polvani et al., 2011). This resulted in the poleward displacement of the SH storm track and led to changes in cloud cover (Grise et al., 2013) and precipitation, not only at the high and mid-latitudes (Polvani et al., 2011; Previdi and Polvani, 2014), but also in the subtropics (Kang et al., 2011). The formation of the ozone hole also affected the Antarctic surface temperatures, with large regional variations in the temperature trend over the continent. Significant warming over the Antarctic Peninsula and Patagonia was reported by Thompson and Solomon (2002). Other consequences of the ozone hole formation include the elevation of the SH polar tropopause (Son et al., 2009; Polvani et al., 2011) and the poleward expansion of the Hadley Cell (Garfinkel et al., 2015; Waugh et al., 2015; Polvani et al., 2011; Min and Son, 2013; Previdi and Polvani, 2014) in austral summer. The wind stress over the Southern Ocean, associated with the westerlies, has also experienced a significant strengthening and poleward shift (Yang et al., 2007; Swart and Fyfe, 2012), with implications for the SH ocean circulation.

Ocean circulation changes due to the formation of the Antarctic ozone hole include the intensification and poleward shift of the SH supergyre, which connects the subtropical Pacific, Atlantic and Indian Oceans (Cai, 2006), an increase in the transport of salty and warm waters from the Indian into the Atlantic Ocean, known as the Agulhas Leakage (Biastoch et al., 2009, 2015; Durgadoo et al., 2013), and changes in the Ekman transport and upwelling in the Southern Ocean (Thompson et al., 2011 and references therein).

As ozone depletion had such profound implications for the SH climate, accurate model simulations of past and future climate change require a correct representation of stratospheric ozone changes and their associated impacts. Multiple lines of evidence suggest that the method used to specify stratospheric ozone in models affects their response to ozone depletion (Gabriel et al., 2007; Crook et al., 2008; Gillett et al., 2009; Waugh et al., 2009; Haase and Matthes, 2019). Ozone concentrations can be calculated interactively (e.g., Haase and Matthes, 2019), as it is the case in chemistry climate models (CCMs) or can be prescribed, either as zonal means or three-dimensionally (3D, e.g., Crook et al., 2008), as monthly-means or at daily resolution. Ozone asymmetries, the temporal resolution of the prescribed ozone field, as well as feedbacks between ozone, temperature, dynamics and transport all impact the way in which changes driven by decreasing ozone concentrations are simulated. This paper investigates how prescribing the ozone field recommended for the Coupled Model Intercomparison Project Phase 6 (CMIP6) affects the atmospheric circulation response to ozone depletion, by drawing a comparison with simulations that calculate the ozone chemistry interactively.

The position of the Antarctic ozone hole is not centered above the South Pole, but varies with that of the polar vortex, being displaced towards the Atlantic sector in the climatological mean (e.g., Grytsai et al., 2007). As a result, the ozone field is characterized by asymmetries in the zonal direction, henceforth referred to as zonal asymmetries in ozone or ozone waves. The effect of zonal asymmetries in ozone was previously investigated for both hemispheres (e.g., Gabriel et al., 2007; Crook et al., 2008). In the Northern Hemisphere winter, zonally asymmetric ozone alters the structure of the stationary wave one, resulting in temperature changes in the stratosphere and mesosphere (Gabriel et al., 2007; Gillett et al., 2009). Ozone waves were also found to affect the number of sudden stratospheric warmings (SSWs), but studies disagree about the sign of the change (Peters et al., 2015; Haase and Matthes, 2019). Peters et al. (2015) reported an increased number of SSWs and a weakening of the Arctic Oscillation between the mid-1980s and mid-1990s in a simulation with specified zonal asymmetries in ozone compared to one in which zonal mean ozone was prescribed. In contrast, Haase and Matthes (2019) found that fewer SSWs occurred between 1955 and 2019 when zonal asymmetries in ozone were prescribed and even less SSWs occurred when the ozone chemistry was calculated interactively. In a recent study, Oehrlein et al. (2020) found no significant difference in the number of midwinter SSWs between their 200-year time-slice simulations with interactive and with prescribed zonally symmetric ozone.

In the SH, the largest zonal asymmetries in ozone occur in spring (Gillett et al., 2009; Waugh et al., 2009), when the stratospheric polar vortex is disturbed by the flux of wave activity from the troposphere and when the ozone hole develops. Model simulations that do not include zonal asymmetries in ozone exhibit a warmer lower stratosphere above Antarctica in austral spring and weaker westerly winds during the decades characterized by strong ozone depletion (Crook et al., 2008; Gillett et al., 2009). The effect of zonal asymmetries in ozone on stratospheric temperature, and hence on the polar vortex, is mediated through changes in stratospheric dynamics and cannot be explained solely by changes in radiative heating associated

with the ozone field (Crook et al., 2008; Li et al., 2016). In addition to differences in the mean state, trends in temperature and in the strength of the stratospheric and tropospheric westerly jets are underestimated in both past (Waugh et al., 2009; Li et al., 2016; Haase et al., 2020) and future (Waugh et al., 2009) simulations that do not include zonal asymmetries in ozone. Furthermore, as the ocean circulation is sensitive to changes in the surface wind stress, it is also affected by the stratospheric zonal asymmetries in ozone. Weaker spring and summer surface westerlies trends in simulations that prescribe zonal mean monthly mean ozone therefore translate into weaker changes in the SH Ekman transport and in the Meridional Overturning Circulation (Li et al., 2016).

Besides zonal asymmetries in ozone, prescribing monthly mean ozone values that are then linearly interpolated to obtain a higher temporal resolution also leads to differences in atmospheric dynamics compared to simulations using interactive chemistry (Sassi et al., 2005; Neely et al., 2014). Linearly interpolating between prescribed monthly ozone values results in an underestimation of ozone depletion compared to interactive chemistry simulations, as the rapid ozone changes during austral spring cannot be fully captured. The weaker ozone hole, in turn, leads to a warmer lower stratosphere and smaller changes in both the stratospheric and the tropospheric westerly winds. Neely et al. (2014) found that these differences greatly diminish if daily ozone is prescribed instead of monthly mean ozone and concluded that the coarse temporal resolution of the prescribed ozone accounts for the majority of the difference in the austral spring stratospheric temperature and the austral summer stratospheric westerly jet between simulations with prescribed and interactive ozone.

Feedbacks between stratospheric ozone, temperature and dynamics can only occur in models that calculate the ozone chemistry interactively, i.e. in CCMs. The importance of such feedbacks in both hemispheres was previously shown in the studies by Haase and Matthes (2019), Haase et al. (2020) and Oehrlein et al. (2020). Changes in temperature caused by ozone depletion, either directly through radiative cooling or indirectly through changes in dynamics, feed back onto ozone concentrations by altering the rate of the catalytic ozone destruction reactions. At the same time, cooling of the polar caps due to ozone loss enhances the meridional temperature gradient in the stratosphere and, as dictated by the thermal wind balance, strengthens the polar vortices. The stronger westerlies, in turn, impact the upward propagation of planetary waves from the troposphere and therefore lead to changes in the BDC, which transports ozone to high latitudes. Changes in stratospheric dynamics due to ozone depletion thus also feed back onto the ozone concentrations. Haase and Matthes (2019) described one such feedback in the Northern Hemisphere spring, during the break-up of the polar vortex. At this time of the year the westerlies are weak and decreasing ozone levels lead to increased planetary wave forcing. This results in dynamical heating and enhanced ozone transport from the low latitudes, both of which lead to an increase in the ozone concentrations, forming a negative feedback loop. This feedback only occurred in the model simulation in which interactive chemistry was used, and not in the simulations in which either zonal mean or three-dimensional ozone was prescribed, showing the importance of calculating the ozone chemistry interactively. A similar feedback also operates in the SH (Lin et al., 2017, Haase et al., 2020). In addition, a positive feedback was reported in the lower stratosphere for the SH by (Haase et al., 2020). A new study investigating historical SAM trends in the CMIP6 models and their drivers found that an indirect effect of increasing greenhouse gases (GHG) on the SAM due to GHG-induced changes in ozone offsets the direct effect of GHG on the SAM (Morgenstern, 2021). The study showed

that models that do not use interactive chemistry therefore overestimate the contribution of the GHG to the historical SAM strengthening.

Previous research conducted using the same model to test the sensitivity of simulations to the method used to represent ozone thus points to climate models that include interactively calculated ozone chemistry as the preferred choice for studies of past and future climate. In contrast, the tropospheric jet's response to ozone depletion is not significantly different between

models with and without ozone chemistry in studies that used different models to assess the sensitivity of the response to how the ozone is imposed (Eyring et al., 2013; Seviour et al., 2017; Son et al., 2018). In the study of Eyring et al. (2013), however, some of the models categorized as including ozone chemistry actually prescribed the ozone field. The difference to the models without chemistry was, in the case of several models, that the ozone field was produced by the interactive chemistry version of the same model. In addition, using different models to evaluate the impact of the method used to impose ozone changes makes

it difficult to assess how other differences between those models, such as the strength of the stratosphere-troposphere coupling, affect the results.

The computational cost of coupled climate models with interactive chemistry is still very high, especially when long climate simulations are needed, as for CMIP6. Therefore, not all climate models participating in CMIP6 use interactive chemistry (Keeble et al., 2020), but instead use atmospheric chemistry data sets obtained from simulations with CCMs. The new atmo-

spheric ozone field recommended for use in CMIP6 (Hegglin et al., 2016) is 3D and has monthly temporal resolution. The issue of smoothing ozone extremes by linearly interpolating from monthly values to the model time step still remains in CMIP6. Additionally, the prescribed ozone field, which was generated by averaging the output of two different CCMs (Keeble et al., 2020), is not consistent with the dynamics of the models to which it is prescribed and, furthermore, feedbacks between ozone, temperature and dynamics cannot occur. Moreover, Hardiman et al. (2019) showed that a mismatch between the tropopause

height present in the prescribed ozone dataset and the tropopause height in the climate model that uses the prescribed ozone dataset can cause erroneous heating rates around the tropopause. These limitations suggest that there are still differences in atmospheric dynamics between climate models using the prescribed CMIP6 ozone and fully interactive CCMs. In this study, we test this hypothesis for the first time by comparing two ensembles of simulations with the new coupled climate model FOCI (Flexible Ocean Climate Infrastructure, Matthes et al., 2020): one ensemble in which the model uses interactive ozone

chemistry and one ensemble in which the CMIP6 ozone is prescribed. We investigate differences in atmospheric dynamics with respect to both the mean state and multi-decadal trends over the second half of the twentieth century. Details about the climate model FOCI and our methodology can be found in Sect. 2. As the increase in anthropogenic GHG was also reported to lead to changes in the SH circulation (Fyfe et al., 1999; Kushner et al., 2001), we first assess the extent to which the formation of the ozone hole and the increase in GHG contribute to the changes simulated in FOCI in Sect. 3 and we verify the model's ability

to simulate the effects of ozone depletion. We then compare the two ensemble simulations and evaluate the performance of the model with prescribed CMIP6 ozone against the interactive chemistry version of the model in Sect 4. Finally, Sect. 5 presents the discussion of the results, together with our conclusion.

## 2 Model description and methodology

### 2.1 Model description and experimental design

The coupled climate model employed in this study is the new Flexible Ocean Climate Infrastructure (FOCI, Matthes et al., 2020). FOCI consists of the high-top atmospheric model ECHAM6.3 (Stevens et al., 2013) coupled to the NEMO3.6 ocean model (Madec and the NEMO team, 2016). Land surface processes and sea ice are simulated by the JSBACH (Brovkin et al., 2009; Reick et al., 2013) and LIM2 (Fichefet and Maqueda, 1997) modules, respectively. We use the T63L95 setting of ECHAM6, corresponding to 95 vertical hybrid sigma-pressure levels up to the model top at 0.01 hPa and approximately 1.8°

by 1.8° horizontal resolution in the atmosphere. The ocean model, in the ORCA05 configuration (Biastoch et al., 2008), has a nominal global resolution of 1/2° and 46 z-levels in the vertical. FOCI has an internally generated Quasi-Biennial Oscillation (QBO) and includes variations in solar activity according to the recommendations of the SOLARIS-HEPPA project (Matthes et al., 2017) for CMIP6. For the interactive chemistry simulations used in this study, chemical processes were simulated using the Model for Ozone and Related Chemical Tracers (MOZART3, Kinnison et al., 2007), implemened in ECHAM6 (ECHAM6-

HAMMOZ, Schultz et al., 2018). A detailed description of FOCI, including the configuration of ECHAM6-HAMMOZ and its chemical mechanism, can be found in the paper by Matthes et al. (2020). A 1500-year long pre-industrial control simulation with FOCI, allowing for the proper spin-up of the model, serves as the starting point for the simulations described below.

Table 1 gives an overview of the simulations used in this study. Three ensembles, each consisting of three simulations differing only in their initial conditions, were conducted in order to distinguish between the effects of ozone depletion and

those of increasing GHG concentrations on the SH climate. The first ensemble (REF) comprises of transient simulations in which surface volume mixing ratios of both GHG and ODS are prescribed and vary as a function of time according to the historical CMIP6 forcing data set (Meinshausen et al., 2017). Therefore, this ensemble captures the combined effects of ozone depletion and GHG increase. In the second ensemble (NoODS), $CO_2$ and $CH_4$ surface volume mixing ratios are prescribed and vary according to the historical forcing, but the ODS follow a perpetual seasonal cycle representative of the 1960 conditions,

computed for each ODS by taking the mean annual cycle between 1955 and 1965. This ensemble was designed to simulate the effects of increasing GHG in the absence of ozone depletion. Here, we use GHG to refer to $CO_2$ and $CH_4$ only, while the other anthropogenic GHG, including $N_2O$, fall under the ODS category. In the third ensemble (NoGHG), the ODS vary according to the historical forcing, while GHG follow a perpetual 1960 seasonal cycle, meaning that there is no increase in GHG past this date. This experimental design allows us to quantify the impact of the formation of the ozone hole by taking

the difference between REF and NoODS and that of climate change by taking the difference between REF and NoGHG. All of these sensitivity simulations use the FOCI configuration that includes interactive chemistry, such that the chemical-radiative-dynamical feedbacks are captured and the ozone field is consistent with the simulated dynamics. Additionally, the high resolution ocean nest INALT10X (Schwarzkopf et al., 2019) was used for these simulations. Therefore, the REF ensemble differs from the INTERACT $O_3$ ensemble discussed below. The INALT10X nest enhances the ocean resolution to 1/10° over

the South Atlantic Ocean, the western part of the Indian Ocean and over the corresponding Southern Ocean sectors, resolving

the mesoscale eddies found in these regions and allowing us to assess, in a follow-up study, the influence of climate change and of ozone depletion on the ocean circulation around the tip of South Africa.

In order to analyze the differences between simulations with interactive ozone chemistry and simulations with prescribed CMIP6 ozone, two further ensembles were performed, each consisting of three simulations differing only in their initial conditions. For the INTERACT $O_3$ ensemble, FOCI was ran in the configuration with interactive chemistry, such that the chemical reactions that are necessary to represent stratospheric chemical processes were included. Therefore, the feedbacks between the stratospheric ozone, temperature and dynamics occur in INTERACT $O_3$ and the simulated ozone field is consistent with the dynamics. A comparison of the ozone field simulated in INTERACT $O_3$ with observations can be found in the work of Matthes et al. (2020). For the FIXED $O_3$ ensemble, the ozone field recommended for CMIP6 (Hegglin et al., 2016) was prescribed. The CMIP6 ozone field was generated by two CCMs and includes solar variations from the SOLARIS-HEPPA project (Matthes et al., 2017). It is a monthly-mean, three-dimensional field and therefore includes zonal asymmetries in ozone. The monthly mean values were linearly interpolated and prescribed at each model time step. The comparison of the INTERACT $O_3$ and FIXED $O_3$ ensembles sheds light on the impact of prescribing the CMIP6 chemistry on the climate simulated by the coupled climate model FOCI.

## 2.2 Observational data

The Integrated Global Radiosonde Archive (IGRA) version 2 (Durre et al., 2006) temperature was used in order to compare the temperature trends simulated by the FOCI ensembles with observational estimates. Data from eleven Antarctic stations located south of 65°S and offering sufficient coverage for the period 1958-2002 was averaged for each day and pressure level up to 30 hPa. As only the South Pole station is located south of 80°S, a trend computed for the entire polar cap is biased towards the lower latitudes and only the trend derived from the spatial average over the 65°S-80°S latitude band is shown in Fig. 10. The full polar cap IGRA temperature trend is sown in the supplement in Fig. S5. In addition, the temperature and wind from the ERA5 reanalysis (Hersbach et al., 2020) was used to further verify the trends obtained from our model.

## 2.3 Methodology

We used the transformed Eulerian mean framework (Andrews et al., 1987) to calculate the residual circulation and its forcing. According to the downward control principle of Haynes et al. (1991), the residual downward velocity $\overline{w^*}$ at a certain level is driven by the wave dissipation at the levels above. The divergence of the Eliassen-Palm (EP) flux, $(a\cos\phi)^{-1}\nabla \cdot F$, gives a measure of the dissipation of resolved waves, where

$$\nabla \cdot F = \frac{1}{a\cos\phi}\frac{\partial(F_\phi\cos\phi)}{\partial\phi} + \frac{\partial F_p}{\partial p} \tag{1}$$

The components of the EP flux are given by

$$F_\phi = -a\cos\phi\,\overline{v'u'} \tag{2}$$

$$F_p = fa\cos\phi\,\frac{\overline{v'\theta'}}{\overline{\theta_p}} \tag{3}$$

The notation is the same as in Andrews et al. (1987): the overbars denote the zonal mean and the primes denote departures from the zonal mean, $a$ is the radius of the Earth, $\phi$ is the latitude, $u$ and $v$ are the zonal and meridional velocity components, respectively, $f$ is the Coriolis parameter, $\theta$ is the potential temperature and $\theta_p$ is the partial derivative of $\theta$ with respect to pressure. The residual vertical velocity was calculated from the streamfunction, as in McLandress and Shepherd (2009):

$$\overline{w^*} = \frac{gH}{pa\cos\phi}\frac{\partial\Psi}{\partial\phi} \tag{4}$$

where the streamfunction is given by

$$\Psi = -\frac{\cos\phi}{g}\int_p^0 \overline{v^*}(\phi,p)\mathrm{d}p \tag{5}$$

and the meridional residual velocity is given by

$$\overline{v^*} = \overline{v} - \frac{\partial}{\partial p}\left(\frac{\overline{v'\theta'}}{\theta_p}\right) \tag{6}$$

with $g$ being the gravitational acceleration and $H$ the scale height taken as 7000 m. The short wave (SW) and long wave (LW) heating rates are part of the standard FOCI output and the total radiative heating rate was obtained by taking the sum of the two. The dynamical heating rate was calculated as the difference between the temperature tendency at daily resolution and the total radiative heating rate.

The SAM was computed as the first empirical orthogonal function (EOF) of the daily, zonal mean geopotential height anomalies at each pressure level, following the method outlined in Gerber et al. (2010). To obtain the geopotential height anomalies, the weighted global mean geopotential height was first subtracted for each day and at each level and latitude. A slowly varying climatology was then removed, to ensure that the resulting SAM index does not exhibit any long-term trend driven by external climate forcing, such that it only reflects internal variability. The slowly varying climatology was obtained by applying a 60-day low pass filter to the geopotential height anomalies from the global mean. Then, timeseries were created for each day of the year and at each location from the filtered anomalies and each timeseries was smoothed using a 30-year low pass filter. The smoothed timeseries were subtracted from the anomalies with respect to the global mean for each respective day and location. The anomalies thus obtained were multiplied by the square root of the cosine of latitude in order to account for the convergence of the meridians towards the poles (North et al., 1982) and only the anomalies for the SH were retained. The first EOF of these anomalies was calculated at each pressure level and the expansion coefficients (principle component timeseries) were obtained by projecting the anomalies onto the first EOF pattern. The expansion coefficients give the SAM index, normalized to have zero mean and unit variance.

The SAM e-folding timescale was computed for each day of the year at each pressure level using the method of Simpson et al. (2011). The autocorrelation function (ACF) was obtained by correlating the timeseries for a particular day of the year with the timeseries for the days lagging and leading it. The ACF was smoothed at each lag and pressure level by applying a Gaussian filter with a full width at half maximum of 42 days over a 181 day window. An exponential function was then fitted to the smoothed ACF up to a lag of 50 days using the least squares method and the SAM timescale was obtained by taking the lag at which the exponential function drops to $1/e$.

Linear trends were calculated over the 1958-2002 period for the analyzed fields at each level and location and the significance
of the trends was assessed based on a Mann-Kendall test. Where differences between simulations are shown, a two-sided $t$-test
was used to test for significance. The significance is always given at the 95% confidence interval. The differences between the
REF and the NoODS or NoGHG ensembles were computed over the 1978-2002 period, as this is the period characterized by
the strongest ozone depletion.

## 3 Impacts of ozone depletion and climate change on the Southern Hemisphere dynamics

The radiative effects of increasing GHG concentrations lead to cooling of the stratosphere and warming of the troposphere,
enhancing the meridional temperature gradient at the tropopause levels, in a similar manner to ozone depletion. While some
older studies argued that rising levels of GHG are the driver of the historical dynamical changes in the SH (Fyfe et al., 1999;
Kushner et al., 2001; Marshall et al., 2004), at present the general consensus is that the formation of the Antarctic ozone hole
is the main cause of these dynamical changes in austral spring and summer and that increasing GHG played only a secondary
role (Arblaster and Meehl, 2006; McLandress et al., 2011; Polvani et al., 2011; Keeble et al., 2014; Previdi and Polvani, 2014;
World Meteorological Organization, 2018). In this section, we separate the effects of ozone depletion from those of increasing
GHG in FOCI and we verify the ability of the model to correctly simulate the dynamical response to ozone loss.

Figure 1 shows the reduction in ozone above the Antarctic polar cap caused by ODS (panel a) together with the accompanying changes in the SW heating rate (panel b). There is a strong decrease in the ozone volume mixing ratio in the lower
stratosphere in austral spring, peaking in October, in agreement with previous studies (Perlwitz et al., 2008; Son et al., 2010;
Polvani et al., 2011; Eyring et al., 2013). This leads to a significant radiative cooling due to decreased absorption of SW radiation (Fig. 1b). An even stronger SW cooling can be seen above 5 hPa between September and April, in line with the results
of Langematz et al. (2003), who found a reduction of the SW heating rate in the upper stratosphere in response to decreasing
ozone concentrations. A significant SW warming appears in December and January between 50 and 10 hPa, related to an
increased ozone mixing ratio. As will be shown later in this section, these latter changes are attributed to a dynamical response
to the spring ozone loss.

Figure 1c shows the ozone changes caused by increasing GHG. There are two regions of statistically significant ozone
increase: the upper stratosphere in austral spring and summer and the region of the ozone hole. The SW heating rate (Fig. 1d)
exhibits warming in response to increased GHG in the same two regions. As the direct effect of GHG on the SW heating rate is
small (Langematz et al., 2003), this warming is likely caused by the higher ozone levels arising in response to the GHG increase,
and not directly by the GHG themselves. Higher levels of GHG lead to increased emission of LW radiation (not shown) and
have a net cooling effect in the stratosphere. In the upper stratosphere, lower temperatures slow down ozone depletion (Haigh
and Pyle, 1982; Jonsson et al., 2004; Stolarski et al., 2010; Chiodo et al., 2018), explaining the simulated increase in ozone.
The ozone increase in the lower stratosphere is more surprising. Here, colder conditions facilitate the formation of PSCs and
are therefore expected to enhance ozone loss. Solomon et al. (2015) showed that a cooling of 2 K results in 30 DU more total
column ozone loss over Antarctica. Therefore, it does not seem likely that the elevated ozone levels are caused by the radiative

effects of GHG. Instead, we find a small but significant enhancement of the downwelling over the polar cap between 50 hPa and 200 hPa in the second half of October, which is associated with increased wave forcing between 20 hPa and 100 hPa (not shown). This suggests that changes in dynamics are responsible for transporting more ozone into the polar lower stratosphere, in agreement with previous studies that linked a GHG-induced acceleration of the BDC to a decrease in lower stratospheric ozone in the tropics and an increase at high latitudes (Dietmüller et al., 2014; Nowack et al., 2015; Chiodo et al., 2018). As a result, the stratospheric ozone depletion is stronger in the absence of increased GHG (NoGHG experiments) than in their presence (REF experiments). The ozone increase related to GHG, about 0.2 ppmv, is, however, small compared to the ozone loss due to ODS, which exceeds 1.4 ppmv in the region of strongest depletion.

The polar cap temperature response to ozone depletion (Fig. 2a) is closely related to the changes in the SW heating rate shown in Figure 1b. A statistically significant cooling occurs in the lower stratosphere in austral spring and in the upper stratosphere in summer. Additionally, there is a warming above the ozone hole in late spring. It should be noted that the maximum temperature increase above the ozone hole occurs about one month earlier than the maximum shortwave warming, hinting to the fact that it is not a direct radiative effect of the increase in ozone. The temperature decreases are a direct response to ozone depletion. The spring cooling in the lower stratosphere represents the well-known signature of the ozone hole. In contrast to the impact of the ozone hole, there is no significant cooling in the polar lower stratosphere due to increased GHG (Fig. 2c). The cooling resulting from enhanced LW emissions is confined to the upper levels of the stratosphere. The lower stratosphere warms in November in response to GHG. At these levels the SW warming (Fig. 1d) due to the elevated ozone concentrations dominates the LW cooling (not shown) due to GHG, resulting in a net radiative warming.

The zonal wind changes associated with ozone depletion (Fig. 2b) and increasing GHG (Fig. 2d) obey the thermal wind balance. The polar night jet accelerates from October onwards as a consequence of the enhanced meridional temperature gradient caused by ozone loss. The maximum acceleration occurs between November and December (Fig. 2b), concomitant with the strongest cooling (Fig. 2a). This westerly acceleration propagates downwards to the tropospheric eddy-driven jet and reaches the surface in November and December. Figure 3a shows the ozone-induced change in the surface zonal wind for these months. The surface westerlies strengthen on their poleward side and weaken on their equatorward side, shifting poleward. This shift is accompanied by changes in sea level pressure (SLP). The pressure over Antarctica drops significantly and the mid-latitude SLP increases in response to ozone depletion (Fig. 3c), signaling a change towards the positive phase of the SAM. All these changes in the SH dynamics simulated in FOCI in response to ozone depletion, both in the stratosphere and in the troposphere are in good agreement with the results of previous studies that isolated the impacts of ozone loss from those of the increase in GHG (Arblaster and Meehl, 2006; McLandress et al., 2010, 2011; Polvani et al., 2011; Keeble et al., 2014), as well as with the trends from observations and the ERA5 reanalysis presented in Sect. 4. This demonstrates that FOCI is able to capture the effects of ozone depletion and is therefore suited to study how prescribing the CMIP6 ozone affects the simulated climate response to ozone loss.

The response of the stratospheric westerlies to higher GHG concentrations is markedly different from that to ozone depletion (Fig. 2b, d). Driven by the warming over the polar cap, the polar night jet weakens in November south of 60°S (supplementary Fig. S1). This change is much weaker compared to that resulting from ozone loss and is confined to the stratosphere. While

GHG do not cause an acceleration of the polar night jet in FOCI, there is a significant positive change in the zonal wind strength centered around 30°S, extending from the top of the eddy-driven jet into the middle stratosphere (supplementary Fig. S1). This westerly change implies a strengthening of the upper flank of the tropospheric jet, in agreement with the findings of McLandress et al. (2010). Figure 3b shows a map of the annual mean GHG-induced changes in the surface zonal winds. We show the annual mean change due to GHG and not the November-December change, as it was the case for ozone depletion because, unlike the effects of ozone loss, the effects of increasing GHG do not exhibit any seasonality. Although the GHG-induced pattern of zonal wind change is similar to that caused by ozone depletion, the changes are much weaker and mostly insignificant. This indicaes that the magnitude of GHG increase was not large enough to induce a strong strengthening or poleward shift of the surface westerly winds. The SLP response to increasing GHG exhibits a significant increase in the mid-latitudes, over the South Pacific Ocean, but the magnitude of this increase is less than a quarter of that due to ozone loss and there is no significant SLP decrease over the polar cap.

Our sensitivity experiments confirm that the changes in the SH polar night jet (Fig. 2b, d) and eddy-driven jet (Fig. 3a, b) during the later part of the twentieth century were mainly driven by ozone depletion. Increasing GHG have played only a minor role, acting to enhance the effect of the ozone hole in the troposphere and to partially counteract the impact of ozone loss on the polar night jet. In addition, we found that the upper stratosphere has cooled significantly and the troposphere has warmed significantly in response to increasing concentrations of GHG.

Having distinguished the contributions of ozone loss and rising GHG levels to the changes in the westerly winds, we now turn our attention to the impacts of ozone depletion on the BDC. Figure 4 shows the November ozone-induced changes in the residual circulation, which is commonly used as a proxy for the BDC. The residual circulation is primarily forced by the dissipation of vertically propagating planetary waves from the troposphere. Therefore, we also present in Figure 4 the changes in eddy heat and momentum fluxes, which reveal the direction of wave propagation, and the changes in the divergence of the EP flux, which measures the wave forcing. The EP flux divergence (Fig. 4c) is characterized by a significant negative change above 10 hPa (stronger convergence) and a significant positive change (weaker convergence) in the lower stratosphere, below 50 hPa. This implies a reduction of wave dissipation in the lower stratosphere and an increase above, suggesting that atmospheric waves propagating from the troposphere reach higher into the stratosphere. This is confirmed by the strengthening of the eddy heat flux above 50 hPa (Fig. 4a). The eddy heat flux is equivalent to the vertical component of the EP flux and gives a measure of the vertical propagation of resolved waves. This strengthening entails increased wave propagation in the middle and upper stratosphere. Similarly, the eddy momentum flux exhibits a negative change, implying increased equatorward wave propagation (Fig. 4b). The ability of the waves to propagate deeper into the stratosphere is related to the strengthening of the polar night jet in response to ozone depletion. Enhanced westerly velocities in November lead to a delay in the breakdown of the polar vortex (e.g., Waugh et al., 1999; Langematz et al., 2003) and sustain wave activity. McLandress et al. (2010) showed that, as a result of ozone depletion, 1) the height of the transition between westerly and easterly velocities has increased, implying that waves can propagate higher at the end of spring, and 2) the date of this transition has been delayed by 10 to 15 days, implying that the period during which waves can penetrate into the stratosphere has been prolonged. As a result, the wave drag due to the dissipation of resolved waves increased in the upper stratosphere and decreased in the lower stratosphere in

November (Fig. 4c), while it increased in the middle stratosphere in December (supplementary Fig. S2), driving similar changes in the residual circulation. In November, the residual meridional velocity (Fig. 4d) shows a significant poleward intensification above 20 hPa and a significant weakening below 50 hPa, in good agreement with the changes in the EP flux divergence. The downwelling over the polar cap is enhanced above 50 hPa (Fig. 4e). Associated with this intensification is a large dynamical warming (Fig. 4f) that increases the temperature above the ozone hole, as shown in Fig. 2a, consistent with the results of Mahlman et al. (1994), Li et al. (2008), Stolarski et al. (2010), Keeble et al. (2014) and Ivy et al. (2016). At the same time, the strengthening of the residual circulation transports more ozone to the polar regions, leading to the increase in ozone seen in December between 50 hPa and 10 hPa in Fig. 1a. The residual vertical velocity in the lower stratosphere is expected to weaken in response to the decreased wave drag seen in Fig. 4c below 50 hPa. Such a weakening is simulated in FOCI at 200 hPa (Fig. 4e), accompanied by a decrease in dynamical heating (Fig. 4f). However, the lower stratospheric change in downwelling is not significant at the 95% confidence interval and the change in the dynamical heating is only partly significant. The decrease in austral spring lower stratosphere downwelling was previously reported by Li et al. (2008), McLandress et al. (2010) and Lubis et al. (2016), while the decrease in dynamical heating was shown by Keeley et al. (2007), Orr et al. (2013) and Lubis et al. (2016). Consistent with our results, McLandress et al. (2010) also attributed their weaker downwelling to reduced wave drag in the austral spring. We note that Figure 4 displays changes averaged for the entire month of November. However, the analysis of Orr et al. (2012, 2013) using 15-day averages showed that, at this time of the year, changes in the lower stratosphere wave driving and dynamical heating due to ozone depletion occur over a shorter time. Therefore, it is likely that our November averaging is applied over periods exhibiting changes of different sign, consequently diminishing the magnitude of the change and rendering it insignificant. At the same time, the large internal variability in FOCI makes it hard to discern this change using fields with higher temporal resolution and more ensemble members would be needed to clearly detect the weakening in downwelling.

The temporal evolution of the ozone-driven changes in wave forcing and, as a result, in the residual circulation can be seen by comparing Fig. 4 with supplementary Fig. S2, which shows the same quantities, but for December. It is clear that there is a downward propagation of the changes in all quantities from November to December. As the polar vortex breaks down at the upper levels, the zonal velocities remain westerly below 50 hPa (contours in Fig. 2b) and are still able to support the remnant wave propagation. Stronger westerlies in the lower stratosphere in December imply enhanced wave dissipation. As a result, the downwelling is accelerated in the lower stratosphere, driving dynamical warming there. These results are consistent with those of McLandress et al. (2010) and explain the reason behind the change in the sign of the residual vertical velocity trends in the lower stratosphere between spring and summer.

Our results clearly show that ozone depletion had a significant influence on the SH BDC in austral spring and summer. FOCI simulates little significant residual circulation changes in the SH due to increasing GHG. Therefore, we conclude that the historical changes in the SH residual circulation over the period of ozone depletion are a consequence of the formation of the ozone hole, in line with the findings of Keeble et al. (2014), Oberländer-Hayn et al. (2015), Polvani et al. (2018), Li et al. (2018), Abalos et al. (2019) and the most recent Scientific Assessment of Ozone Depletion (World Meteorological Organization, 2018). Consistent with these studies, FOCI simulates a strengthening of the SH residual circulation in response

to enhanced wave forcing at the end of the spring and the beginning of summer. At the same time, a weakening of the spring lower stratosphere residual circulation is simulated, as found by the few studies that investigated springtime changes in the BDC (Li et al., 2008; McLandress et al., 2010; Lubis et al., 2016). The good agreement with previous studies demonstrates that the interactive chemistry configuration of FOCI adequately simulates the impact of ozone depletion on the residual circulation.

## 4  Effects of prescribing the CMIP6 ozone field

We aim to understand how prescribing the ozone field recommended for CMIP6 affects the SH atmospheric circulation response to ozone depletion. To this end, we compare an ensemble of simulations using prescribed CMIP6 ozone with an ensemble of simulations that use fully interactive chemistry. The use of an ozone field that differs from that internally simulated by the model and that is not consistent with the model dynamics, the lack chemical-radiative-dynamical feedbacks and the temporal interpolation from the monthly prescribed values to the model time step can all lead to differences between the two ensembles. With a view on these deficiencies, we begin by analyzing the differences in the mean state in Sect. 4.1 and we then compare the simulated SH variability in Sect. 4.2 and the persistence of the SAM in Sect. 4.3.

### 4.1  Effects on the mean state

Figure 5 shows the difference between INTERACT $O_3$ and FIXED $O_3$ in October average ozone and November average temperature, geopotential height and zonal wind at 70 hPa. The CMIP6 ozone field was used for FIXED $O_3$. FOCI simulates significantly lower ozone levels above the Antarctic Peninsula and the Bellingshausen Sea in October compared to the CMIP6 ozone (Fig. 5a). As a consequence, the November temperature (Fig. 5b) and the geopotential height (Fig. 5c) are also lower in this region in INTERACT $O_3$ compared to FIXED $O_3$. We note that the pattern of the temperature difference between INTERACT $O_3$ and FIXED $O_3$ is markedly different to the pattern reported by Crook et al. (2008) and Gillett et al. (2009), which arises due to zonal asymmetries in ozone. The CMIP6 ozone field prescribed in FIXED $O_3$ includes ozone asymmetries and their effects are therefore captured in FIXED $O_3$. Despite this, spatial temperature and geopotential height differences still remain between simulations with prescribed ozone asymmetries and simulations with fully interactive ozone chemistry, because the prescribed ozone field differs from the simulated one and it is not consistent with the simulated dynamics.

The differences between the FOCI and the CMIP6 ozone fields are not confined just to the ozone hole itself. Outside of the polar vortex, INTERACT $O_3$ exhibits significantly higher ozone levels at all longitudes. The difference in ozone maximizes in the eastern hemisphere, as the polar vortex, and hence the ozone hole, is not centered over the pole, but displaced towards the Atlantic Ocean and South America (contours in Fig. 5a). This significant positive difference was found in the mid- to high-latitudes of both hemispheres and in all seasons (not shown). We hypothesize that it is the result of a stronger BDC in FOCI compared to the models used to generate the CMIP6 ozone field, leading to increased ozone transport from the tropics. Associated with the higher ozone levels, the November mid-latitude temperature and geopotential height are also elevated in INTERACT $O_3$ compared to FIXED $O_3$. In the Atlantic and Indian sectors, the higher temperature outside of the polar vortex in INTERACT $O_3$ enhances the meridional pressure gradient between the polar low and the mid-latitude high. In the Pacific

sector, the meridional pressure gradient is stronger in INTERACT $O_3$ due to the lower temperature above West Antarctica and the Bellingshausen Sea. As a result, the November polar night jet is circumpolarly stronger in INTERACT $O_3$ compared to FIXED $O_3$ (Fig. 5d).

To better understand the cause of the lower ozone levels above the Antarctic Peninsula and the Bellingshausen Sea in INTERACT $O_3$, Fig. 6c shows the October average ozone anomalies from the zonal mean in INTERACT $O_3$ at 70 hPa, as well the difference to FIXED $O_3$. A zonal wavenumber one pattern is clearly visible, with the ridge at the edge of Antarctica towards New Zealand and the trough over the tip of the Antarctic Peninsula. The ozone wave simulated in FOCI is consistent with that inferred from satellite observations by Lin et al. (2009) and Grytsai et al. (2007), from reanalyses by Crook et al. (2008) and with that simulated by Gillett et al. (2009). This wave pattern confirms that the simulated ozone hole is not centered on the south pole. While the CMIP6 ozone hole is also displaced from the pole, its location and extent is not the same as that simulated by FOCI (compare contours in Fig. 7a and d). The difference shown in Fig. 6c reveals that, on the one hand, the trough of the wave is shifted towards South America and reaches deeper into the Pacific sector in INTERACT $O_3$. On the other hand, the amplitude of the wave is significantly greater in INTERACT $O_3$ (compare also contours in Fig. 8a and b). Figure 6c thus demonstrates that the prescribed CMIP6 ozone field is not spatially consistent with the polar vortex simulated in FOCI.

Figures 6a and 6b show the time evolution of the ozone wave averaged between 60°S and 70°S for the month of October for INTERACT $O_3$ and FIXED $O_3$, respectively. Despite considerable interannual variability, the westward shift of the wave in INTERACT $O_3$ compared to that in FIXED $O_3$ is clearly discernable. This can be attributed to different evolutions of the wave trough in the two ensembles. Figures 6f and 6g show the timeseries of the longitudes at which the ozone maximum occurs within the ridge of the wave and at which the ozone minimum occurs within the trough of the wave, respectively, together with their corresponding trends. While the FIXED $O_3$ trough exhibits a significant eastward shift of $14.75 \pm 4.45$ ° $\mathrm{dec}^{-1}$, the INTERACT $O_3$ trough does not exhibit a significant shift. In the time mean, this results in the INTERACT $O_3$ ozone trough being located more to the west than the FIXED $O_3$ trough. The ridge of the wave shifts eastwards in both ensembles (Fig. 6f). Although the magnitude of the shift is stronger in FIXED $O_3$, the trends are not significantly different, as they fall within each other's 95% confidence intervals. Both ozone fields exhibit a deepening of the wave pattern over time, in particular in the 1980s, as the ozone hole becomes stronger, in agreement with the increase in the ozone wave amplitude reported by Grytsai et al. (2007) and Crook et al. (2008). In addition, the slow eastward shift of the wave with time is consistent with the phase shift based on temperature observations reported by Lin et al. (2009). An eastwards shift of the ozone zonal wave one was also inferred from satellite observations by Grytsai et al. (2007), accompanied by a westward shifting zonal wave number two. Due to the superposition of the two wave numbers, only the through of the wave sum shifted eastward, while the ridge remained stationary. Dennison et al. (2017) fitted an ellipse to the ozone hole and showed that the central longitude of the ellipse moves westward as the ozone is depleted. Due to the use of different methods, it is not straightforward to compare the results of Dennison et al. (2017) with the results presented here, or in the studies of Lin et al. (2009) and Grytsai et al. (2007). In our analysis, both the through and the peak exhibit a phase shift in FIXED $O_3$, but only the ridge exhibits a eastward shift in INTERACT $O_3$. This results in the shift being stronger in FIXED $O_3$ (Fig. 6b) and less evident in INTERACT $O_3$ (Fig. 6a).

The consequences of prescribing an ozone field that is not consistent with the model dynamics are depicted in Fig. 6d and e. The wave one pattern can be seen in both temperature and geopotential height anomalies from the zonal means in October. In both fields there are significant differences between INTERACT $O_3$ and FIXED $O_3$. Consistent with the ozone anomalies, the temperature trough is shifted westward in INTERACT $O_3$ and the amplitude of the wave is stronger. The radiative effects of a prescribed ozone hole that is not collocated with the polar vortex appear to alter the location of the wave, while the weaker amplitude of the prescribed ozone wave affects the amplitude of the temperature response. The westward shift of the INTERACT $O_3$ wave is also seen in the geopotential height field (Fig. 6e). We thus conclude that prescribing the CMIP6 ozone field, which is not consistent with the model dynamics in general and with the simulated stratospheric polar vortex in particular, gives rise to significant differences in the spatial structure and the amplitude of the springtime lower stratospheric wave one, as well as to significant differences in the springtime climatological strength of the polar night jet. Although we presented here evidence for the 70 hPa level, the results hold true for levels throughout the lower and middle stratosphere.

## 4.2 Effects on the simulated Southern Hemisphere variability

### 4.2.1 Temperature

We now turn our attention to the differences in the trends simulated by the two ensembles in response to ozone depletion. First, we compare the October and November 100 hPa ozone trends between INTERACT $O_3$ and FIXED $O_3$ (Fig. 7a, d, g, j), as they are at the root of the polar stratospheric temperature changes. We also examine the eddy contribution to the October trends, i.e. the trends in the anomalies from the zonal mean (Fig. 8a, b). The FIXED $O_3$ ozone is given by the prescribed CMIP6 ozone field.

The total spatial extent of ozone depletion is greater in INTERACT $O_3$ compared to FIXED $O_3$, but in both cases, in October, the ozone trend maximizes over East Antarctica (Fig. 7a, d). In contrast, in November the FIXED $O_3$ trend maximum covers most of the Antarctic continent, while the INTERACT $O_3$ trend maximum is located over East Antarctica (Fig. 7g, j). The maximum ozone depletion is stronger in FIXED $O_3$ than in INTERACT $O_3$ during both October and November (Fig. 7a, d, g, j). The trends in ozone anomalies from the zonal mean exhibit a wave one structure, but the wave is shifted eastwards compared to its climatological position in both ensembles (Fig. 8a, b). This implies a progressive eastward migration of the wave structure over time and an increase in the amplitude of the wave, in agreement with Fig. 6. The locations of the October trend minimum over the Atlantic sector and of the trend maximum over the Pacific sector are consistent with observations (Lin et al., 2009).

There are notable differences between the eddy component of the ozone trends in INTERACT $O_3$ and FIXED $O_3$. The position of the trough is farther east in FIXED $O_3$, while the ridge extends more towards the Drake Passage. In INTERACT $O_3$, the trend trough is strong and significant over the Drake Passage, but in FIXED $O_3$ it has the opposite sign and it is not significant. The pattern of the trend trough in INTERACT $O_3$, which extends into the Pacific, to the west of the time mean trough, explains the lack of an eastward shift of the ozone wave trough in this ensemble, as shown in Fig. 6g. In addition, the ridge of the trend wave is stronger in INTERACT $O_3$ than in FIXED $O_3$. These differences in the trends of the ozone fields passed to the radiation scheme in the two ensembles translate into differences in temperature trends between them. Similarly

to the trend in ozone, the eddy component of the temperature trend exhibits a wave one structure which is shifted eastward in FIXED $O_3$ compared to INTERACT $O_3$, albeit less than in the case of ozone (Fig. 8c, d). In agreement with the stronger ozone trend ridge in INTERACT $O_3$, the temperature trend ridge is also stronger than in FIXED $O_3$, with the latter not being statistically significant. The eastward displacement of the temperature trend pattern in FIXED $O_3$ shows that prescribing an ozone field inconsistent with the dynamics of the model to which it is prescribed alters the spatial structure of the temperature trend. We also note that the ozone and temperature trend wave patterns are not properly collocated in FIXED $O_3$, as it is the case in INTERACT $O_3$, further highlighting the discrepancies between the prescribed ozone field and the simulated dynamics.

Even larger differences appear when the full polar temperature trends are compared (Fig. 7b, e, h, k). Despite its stronger ozone trend (Fig. 7a, d, g, j), FIXED $O_3$ displays a considerably weaker temperature trend over the entire Antarctic continent. This is true for each of the individual ensemble members in FIXED $O_3$ and INTERACT $O_3$ (Fig. 10c and supplementary Fig. S3), indicating that the spring lower stratospheric temperature trends are significantly different in simulations with prescribed CMIP6 ozone and with interactive ozone chemistry. The trend in the SW heating rate (Fig. 10b and supplementary Fig. S4) is stronger in FIXED $O_3$ than in INTERACT $O_3$, consistent with the trends in the respective ozone fields (Fig. 7a, d, g, j). The net radiative contribution to the temperature trend is weaker in both experiments due to the partial cancelation of the SW cooling and LW warming trends (not shown). Nevertheless, the stronger SW heating trend in FIXED $O_3$ demonstrates that the temperature trend differences between INTERACT $O_3$ and FIXED $O_3$ cannot be explained by the differences in the trends of the imposed ozone fields, nor can they be explained by the linear interpolation of the monthly CMIP6 ozone field to the model time step, as it was the case in the studies by Sassi et al. (2005) and Neely et al. (2014). If that were the case, the smaller ozone extremes resulting from the interpolation would also reduce the FIXED $O_3$ SW heating rate trend, not just the temperature trend. Anomalies from the daily climatology of the polar cap SW heating rate (supplementary Fig. S7) show more strong positive extremes in the 1960s and 1970s and more strong negative extremes after 1990 in FIXED $O_3$ than in INTERACT $O_3$, further confirming that differences in the SW heating rate variations cannot explain the stronger temperature trend in INTERACT $O_3$. The key to the different temperature trends in INTERACT $O_3$ and FIXED $O_3$ lies instead in the dynamical heating rate trends (Fig. 7c, f, i, l). In INTERACT $O_3$, there is a strong and significant dynamical cooling trend in October over the majority of the Antarctic continent, peaking and becoming significant over East Antarctica (Fig. 7f). This dynamical cooling trend becomes stronger in November (Fig. 7l) and moves away from the Pacific, while a dynamical warming trend appears south of Australia and New Zealand. The dynamical cooling trend greatly amplifies the lower stratospheric temperature trend due to the radiative heating rate in INTERACT $O_3$. In FIXED $O_3$, the October dynamical cooling trend over East Antarctica is much weaker and it is not significant, while a warming trend is visible over parts of the continent (Fig. 7c). A strong and significant warming trend appears over the Ross Sea during November in FIXED $O_3$. Consequently, the dynamical heating rate in FIXED $O_3$ brings a negligible contribution to the temperature trend in October and it acts to offset the radiative cooling trend in November, explaining the considerably weaker cooling in this ensemble. The strongest dynamical cooling trend in FIXED $O_3$ occurs over the Atlantic sector of the Southern Ocean during October. The dynamical heating rate is closely related to the polar downwelling, which is, in turn, controlled by the wave dissipation above. Therefore, the different October and November dynamical cooling trend patterns in Figs. 7c, i and f, l have two implications: 1) that wave dissipation is suppressed and 2) that

the downwelling weakens in completely different regions in the two ensembles. The cause of these spatial discrepancies is the spatial inconsistency between the prescribed ozone hole and the simulated polar vortex in FIXED $O_3$. The stronger temperature trend in INTERACT $O_3$ compared to FIXED $O_3$ can be explained by the different lower stratospheric dynamical responses to ozone depletion in the two ensembles.

The temperature trends are weaker in FIXED $O_3$ throughout the lower stratosphere (Fig. 9a, b). The spread among the individual ensemble members is shown is supplementary Fig. S5 and all three INTERACT $O_3$ members exhibit stronger cooling in the lower stratosphere during spring compared to all three FIXED $O_3$ members, confirming that the difference between the ensemble mean trends is significant. Figure 9c shows the seasonal evolution of the temperature trends from the ERA5 reanalysis. Like INTERACT $O_3$ and FIXED $O_3$, ERA5 exhibits the maximum cooling in November at 100 hPa. The

maximum cooling is stronger in ERA5 than in FIXED $O_3$ and it is closer in magnitude to the INTERACT $O_3$ cooling, but its vertical extent, as well as its duration, are reduced compared to INTERACT $O_3$. Figure 10c shows the November 100 hPa temperature trend for the individual members of INTERACT $O_3$ and FIXED $O_3$, as well as the trends from the ERA5 reanalysis and the IGRA radiosonde observations. As the IGRA temperature data contains only one station located south of $80°$S, the temperature trend in Fig. 10c is given for $65°$S-$80°$S, in order not to bias the IGRA trend towards the lower latitudes.

The trends for the entire polar cap can be compared in Figs. 9 and S5. Despite considerable spread among the individual ensemble members, all INTERACT $O_3$ members exhibit stronger temperature trends compared to the FIXED $O_3$ members. This contrasts the trends in ozone (Fig. 10a) and SW heating rate (Fig. 10b), which are stronger in FIXED $O_3$, in agreement with Figs. 7 and S4. The lack of spread in the SW heating rate trend among the FIXED $O_3$ members seen in Fig. 10b is a consequence of imposing the same ozone field in each of these members. The ERA5 and IGRA temperature trends fall in the

lower end of the INTERACT $O_3$ trend range, but are stronger than the trends in all of the FIXED $O_3$ members (Fig. 10c). The apparent disagreement between the weaker ozone and SW heating rate trends (Fig. 10a, b) and the stronger temperature trend (Fig. 10c) in INTERACT $O_3$ compared to FIXED $O_3$ can be explained by the dynamical cooling trend present in the first ensemble (Fig. 13e). Comparing the spring lower stratospheric temperature trends in INTERACT $O_3$ and FIXED $O_3$ with trends from the ERA5 reanalysis and from the observations shows that FIXED $O_3$ tends to underestimate the ozone-induced

cooling, while INTERACT $O_3$ tends to overestimate it, although individual ensemble members can simulate trends that are close to those observed.

    The lack of observations in the middle and upper stratosphere, particularly before the satellite era, cast doubt on the reliability of the ERA5 temperature trends at these levels (Fig 9c). The very strong cooling seen between April and September above 20 hPa is likely spurious. A significant warming trend can be seen between 50 hPa and 10 hPa above the ozone hole during

November and December, in agreement with the FOCI simulations. This warming is weaker in FIXED $O_3$ and ERA5 than in INTERACT $O_3$. As shown in Sect. 3, this warming is attributed to changes in the strength of the BDC. Its different magnitude between INTERACT $O_3$ and FIXED $O_3$ thus points to the role played by the dynamics in setting the different responses to ozone depletion in the two model configurations. In contrast, the magnitude of the cooling at the stratopause levels in winter and fall is similar between INTERACT $O_3$ and FIXED $O_3$, as this cooling is a purely radiative effect of ozone loss and increase

in GHG (Sect. 3).

### 4.2.2 Zonal winds

The magnitude of the stratospheric circumpolar westerly winds trends in INTERACT $O_3$ and FIXED $O_3$ is consistent with the magnitude of the temperature trends. A stronger lower stratosphere cooling in INTERACT $O_3$ implies a more pronounced enhancement of the meridional temperature gradient and hence a stronger acceleration of the westerly winds (Fig. 9d, e). All INTERACT $O_3$ members simulate a stronger acceleration of the stratospheric westerlies compared to all FIXED $O_3$ members (Figs. 10d and S6), confirming that the trends are significantly different between the two ensembles. Similar to the temperature trends, the magnitude of the maximum stratospheric westerlies acceleration in ERA5 lies between that in the INTERACT $O_3$ and the FIXED $O_3$ ensemble means (Figs. 9d, e, f). However, unlike the temperature trends, the ERA5 wind trend falls within the range of the FIXED $O_3$ trends and is smaller than all the INTERACT $O_3$ trends (Fig. 10d). The zonal wind trend in INTERACT $O_3$ is not only stronger, but it also occurs about half a month earlier than in FIXED $O_3$ and ERA5, from the end of September onwards. Although there are pronounced differences between the ERA5 and both the INTERACT $O_3$ and the FIXED $O_3$ upper stratospheric wind trends, the reliability of the ERA5 trends is questionable due to a lack of observations at these levels.

The timing of the downward propagation of the zonal wind trends to the troposphere also differs between the two ensembles. Figure 11 shows the seasonal cycle of the 850 hPa zonal wind trends as a function of latitude. A significant poleward intensification of the tropospheric westerlies can be seen from late November through January in INTERACT $O_3$, accompanied by a deceleration on the equatorward side. In contrast, the westerly winds in FIXED $O_3$ exhibit significant trends on both flanks only starting in late December and the trends remain significant throughout February. The timing of the ERA5 850 hPa westerlies strengthening agrees better with FIXED $O_3$, although there is a large number of days with insignificant trends between December and February (Fig. 11e). The weakening of the westerlies on their equatorward side is not significant in the zonal mean in ERA5. Figure 11 also shows maps of the zonal wind trends at 850 hPa averaged over December and January, the two months in which both ensembles exhibit significant trends in the troposphere. The trends are a circumpolar feature and are characterized by strengthening on the poleward side and a weakening on the equatorward side of the tropospheric jet, implying that the jet has shifted poleward in both ensembles. The maximum magnitude of the poleward strengthening is similar in INTERACT $O_3$ and FIXED $O_3$, but there are clear spatial differences between the two ensembles. Figure 10e shows the shift in the latitude of the zonal mean jet maximum for December and January separately. The poleward shift in the ensemble mean westerlies is significant in December for INTERACT $O_3$ and in both December and January for FIXED $O_3$, although not all ensemble members exhibit a significant shift. ERA5 exhibits a significant poleward shift of the westerlies only in January. The positive and the negative wind trends in ERA5 are less zonal than in FOCI (Fig.11f), with the positive trend reaching equatorward of 40°S in the Pacific sector. This could partly explain the lack of a significant equatorward weakening of the westerlies when the zonal mean trends are considered (Fig.11e), as trends of opposite sign cancel each other out in the zonal mean.

Figure 10f shows the December and January westerly wind trends for the latitude band where the trend is positive (45°S-70°S). We chose to average the winds over this latitude band as opposed to taking the trends at the latitude where the westerlies maximize because the westerlies maximum is located in the transition zone between positive and negative trends and therefore

exhibits weak trends that are not significant (Fig. 11). FIXED $O_3$ exhibits trends of comparable magnitude in both months. Although the FIXED $O_3$ ensemble mean trend is significant, only one member shows a significant trend in December, while two members show significant trends in January. The ensemble mean exhibits significant trends when the individual members do not because the ensemble averaging is reducing the interannual variability, which is large in the individual members. All INTERACT $O_3$ members exhibit a significant strengthening of the westerlies in December, but this ensemble does not show a significant zonal mean trend in January. The zonal mean ERA5 westerlies strengthening is not significant in either December or January, although certain regions show significant trends, as seen in Fig. 11f.

The tropospheric westerly jet trends in INTERACT $O_3$ and FIXED $O_3$ cannot be clearly differentiated, as it is the case for the stratospheric jet. Both the poleward shift of the jet as well as its strengthening appear stronger in INTERACT $O_3$ in December, but the ranges of the trends given by the individual ensemble members overlap. In contrast, in January the INTERACT $O_3$ trends tend to be weaker than the FIXED $O_3$ trends and they are not significant, with one member showing trends in the opposite direction. However, there is a large overlap between the range of the trends in the two ensembles. The maps of the December-January averaged trends (Fig. 11b, d) show maxima of similar magnitude, albeit at different locations. Therefore, we conclude that the differences in the tropospheric westerly jet trends in INTERACT $O_3$ and FIXED $O_3$ are within the range of internal variability, in agreement with the results of Eyring et al. (2013), Seviour et al. (2017) and Son et al. (2018).

### 4.2.3 Mechanisms for the different responses to ozone depletion

To obtain a clear understanding of the dynamical mechanisms responsible for the temperature and zonal wind trend differences between INTERACT $O_3$ and FIXED $O_3$, we investigate the trends in the residual circulation and its drivers. Figure 12 shows vertical profiles of the November trend in the eddy heat flux, the divergence of the EP flux, the vertical and the meridional components of the residual circulation. Trends for October and December have additionally been examined, but are not shown. The October and November dynamical heating rate trends are illustrated in supplementary Fig. S8. Figure 13 shows the spread of the November trends in these fields between the individual ensemble members. We selected the pressure levels where the trends tend to maximize. In choosing the pressure levels, we took care that the trends in the divergence of the EP flux and in the meridional residual velocity are given at a height above the trends in the vertical residual velocity, as the downward control principle (Haynes et al., 1991) states that the vertical residual velocity is driven by wave dissipation above. In the lower stratosphere, we show dynamical heating and vertical meridional velocity trends at 100 hPa, where the temperature trends maximize.

We first focus on the different onset dates of the westerly wind trends. A significant positive trend can be seen in the middle and upper stratosphere in INTERACT $O_3$ already at the beginning of October (Fig. 9d), but it only appears in the second half of October in FIXED $O_3$ (Fig. 9e). There is also a significant cooling visible in the polar cap middle stratosphere at the beginning of October in INTERACT $O_3$ (Fig. 9a). As this trend occurs above the ozone hole, it is likely the result of dynamical rather than radiative changes. At the beginning of October, the westerly velocities in the polar night jet (contours in Fig. 9d, e) are close to the critical velocity for Rossby wave propagation (Charney and Drazin, 1961). The positive trend (Fig. 9d) is enhancing the winds past this critical velocity and it inhibits wave propagation in INTERACT $O_3$. A significant decrease in

wave dissipation in the lower and middle stratosphere occurs in October in INTERACT O$_3$, but not in FIXED O$_3$ (not shown). This October reduction in wave forcing in INTERACT O$_3$ is consistent with the findings of McLandress et al. (2010), who also used a CCM in their study. The reduced wave forcing has two effects: first, less easterly momentum is deposited through wave dissipation, further enhancing the westerly winds. Second, the polar cap downwelling in the lower stratosphere is weakened due to the decreased wave forcing above, leading to a significant dynamical cooling. A significant October polar cap dynamical cooling trend occurs in the lower stratosphere only in INTERACT O$_3$ and it is absent in FIXED O$_3$ (Figs.7c, f and S8a, b). The lack of a dynamical cooling trend in FIXED O$_3$ is explained by a prescribed ozone hole that is not consistent with the dynamics, which alters the propagation of planetary waves, as demonstrated above. The dynamical cooling in INTERACT O$_3$ reinforces the negative temperature trend due to the radiative effects of ozone loss and further strengthens the westerly winds via the thermal wind balance, resulting in stronger temperature and zonal wind trends compared to FIXED O$_3$.

In November, the polar vortex breaks down and climatological easterly winds occur in the upper and middle stratosphere (contours in Fig. 9d, e). The positive zonal wind trend prolongs the life of the polar vortex, delaying its breakdown (e.g. McLandress et al., 2010) and allowing planetary waves to propagate higher into the stratosphere, as shown by the negative trend in the eddy heat flux (Fig. 12a, e). Wave dissipation is enhanced in the middle and upper stratosphere and decreased in the lower stratosphere in both FIXED O$_3$ and INTERACT O$_3$ (Fig. 12b, f), but the magnitude of the INTERACT O$_3$ trends is larger. As shown in Fig. 13a and b, the upper stratospheric negative trends in both the eddy heat flux and the divergence of the EP flux are stronger in all the INTERACT O$_3$ simulations than in all the FIXED O$_3$ simulations. The positive EP flux divergence trends in the lower stratosphere are not as clearly separated between the two ensembles, but two of the INTERACT O$_3$ simulations exhibit significant trends, while none of the FIXED O$_3$ simulations, nor the ensemble mean do. The region of enhanced wave dissipation extends farther down in FIXED O$_3$, confining the region of reduced wave dissipation to the levels below 100 hPa. This has important implications for the residual circulation trends. In FIXED O$_3$, the negative trend in the polar cap downwelling reaches all the way into the lower stratosphere (Fig. 12c). The accompanying polar cap dynamical warming trend (Figs. 7i and S8d) partly offsets the radiative cooling trend due to ozone depletion, resulting in the weaker temperature trend seen in Fig. 9b. In contrast, the polar downwelling in INTERACT O$_3$ strengthens only above 50 hPa and exhibits an insignificant weakening below (Fig. 12g). The meridional residual velocity follows a similar pattern, strengthening in the upper stratosphere and weakening in the lower stratosphere (Fig. 12h). There is a clear distinction between the INTERACT O$_3$ and FIXED O$_3$ ensembles regarding the magnitude of the residual circulation and dynamical heating trends in the middle and upper stratosphere, with INTERACT O$_3$ exhibiting a stronger intensification of the residual circulation (Fig. 13c, d, e). In the lower stratosphere, only the dynamical heating trends can be separated. Two out of three INTERACT O$_3$ members simulate a significant cooling, while the FIXED O$_3$ ensemble mean simulates a significant warming, despite the fact that the warming in the individual members is not significant (Fig 13e). This warming is consistent with the enhanced polar cap downwelling simulated by the FIXED O$_3$ simulations (Fig 13d). In contrast to the weakening of the meridional residual velocity and to the dynamical warming simulated by INTERACT O$_3$, this ensemble does not exhibit a weakening of the polar cap downwelling in the lower stratosphere. While this is surprising, it is possible that the zonally asymmetric nature of the polar downwelling results in trends of opposite signs canceling each other out when averaged over the polar cap. The constraints of the zonal mean

fields obtained trough the transformed Eulerian mean framework does not allow us to investigate this issue. Nevertheless, the dynamical changes in the lower stratosphere drive a dynamical cooling trend over the polar cap (Figs. 7l, 13e and S8c) that, as in October, adds to the radiative cooling trend due to ozone depletion to produce a stronger negative temperature trend in INTERACT $O_3$. The dynamical warming in the middle and upper polar stratosphere is also stronger in INTERACT $O_3$, explaining the stronger warming trend in Fig. 9a.

The residual circulation in the lower stratosphere thus exhibits a different response to the formation of the ozone hole in FIXED $O_3$ compared to INTERACT $O_3$. These different residual circulation changes are the consequence of the fact that the prescribed ozone hole is not consistent with the simulated dynamics. They are at the core of the stratospheric temperature and westerly winds differences between the two ensembles, as they drive the dynamical heating contribution to the temperature changes.

### 4.3  Effects on the SAM timescale

Dennison et al. (2015) showed that ozone depletion increases the persistence of the SAM in the stratosphere. This raises the question of whether prescribing the CMIP6 ozone affects the impact of the ozone hole on the SAM persistence. Figure 14 shows the seasonal evolution of the ensemble mean SAM timescale for INTERACT $O_3$ and FIXED $O_3$. The SAM timescale was first computed for the individual ensemble members and then averaged to obtain the ensemble means. While both INTERACT $O_3$ and FIXED $O_3$ capture the peak SAM persistence in austral spring and summer (Gerber et al., 2008, 2010), there are marked differences between the timescales in the two ensembles. The middle stratosphere SAM is more persistent in INTERACT $O_3$ between August and January, bringing the amplitude and the seasonal structure of the SAM timescale in INTERACT $O_3$ closer to that exhibited in reanalyses (compare Fig. 14 with Fig. 8b in the study by Gerber et al., 2010). In contrast, FIXED $O_3$ is characterized by a more vertical structure of the maximum persistence, focused in December, and, overall, by a shorter timescale. The inconsistent evolution of the ozone hole and the polar vortex in FIXED $O_3$ dampens the dynamical response to ozone depletion and reduces the SAM timescale. At the same time, the chemical-dynamical feedbacks present only in INTERACT $O_3$ reinforce the ozone-induced anomalies and lead to more persistent polar vortex anomalies, reflected in the longer SAM timescale in this ensemble. In a parallel study conducted with a different CCM by Haase et al. (2020), an ensemble of experiments with prescribed daily zonally asymmetric ozone consistent with the dynamics still exhibits shorter SAM timescales compared to the interactive chemistry experiment. This highlights the role of the feedbacks with the ozone chemistry in prolonging the SAM persistence. The same study additionally shows that the effect of zonal asymmetries in ozone is also to enhance the SAM timescale.

In the troposphere, both INTERACT $O_3$ and FIXED $O_3$ overestimate the SAM timescale and exhibit a peak persistence that is too broad when compared to the SAM timescale derived from reanalyses by Gerber et al. (2010) or Simpson et al. (2011). This is a common problem of climate models reported by Gerber et al. (2008) for the CMIP3 models and by Gerber et al. (2010) for CCMs and it appears to persist in the current generation of climate models. Simpson et al. (2011) attempted to identify the sources of the exaggerated tropospheric SAM persistence and found that the stratospheric SAM persistence contributes about half of the bias, while the other half is of tropospheric origin. The authors cite an equatorward-biased position of the

tropospheric jet as a possible reason for the tropospheric contribution to the overestimation of the SAM timescale. This also holds true in our model, since the tropospheric jet in FOCI is located closer to the equator compared to observations, as it is common in climate models (Swart and Fyfe, 2012). Overall, the use of interactive chemistry improves the representation of the stratospheric SAM timescale in FOCI, but the tropospheric issues common to climate models still remain.

## 5 Discussion and conclusion

The ocean-atmosphere coupled climate model FOCI was used in this study to 1) separate the effects of ozone depletion on the SH temperature and dynamics from those arising from increasing concentrations of GHG and 2) compare these effects between an ensemble of simulations where the CMIP6 ozone field is prescribed and an ensemble of simulations that use interactive chemistry. We found the following:

- The formation of the Antarctic ozone hole is the primary driver of the dynamical changes in the atmosphere that occurred in the SH spring and summer during the last decades. These changes comprise of an acceleration of the polar night jet, which propagated to the surface, where the surface westerlies shifted poleward and the SAM shifted toward its high index polarity.

- The increase in anthropogenic GHG partly offset the November polar night jet response to ozone depletion in the stratosphere. At the surface, it resulted in a similar response of the westerlies and of the SAM, but this response was much weaker compared to that driven by ozone loss and it is generally not statistically significant.

- Ozone depletion drove an intensification of the SH BDC in the middle stratosphere in November and in the lower stratosphere in December. In contrast, a decrease in wave dissipation driven by ozone loss led to the weakening of the residual circulation in the lower stratosphere in November.

- The CMIP6 ozone field is not consistent with the dynamics simulated by FOCI and the ozone hole is not collocated with the polar vortex. Consequently, the austral spring zonal wavenumber one is weaker in FIXED $O_3$ and shifted eastwards compared to INTERACT $O_3$. The austral spring climatological polar night jet is weaker in FIXED $O_3$ than in INTERACT $O_3$.

- The dynamical response to ozone depletion enhanced the radiatively-driven austral spring lower stratospheric cooling. A novel result of this study is that this effect only occurs in INTERACT $O_3$ and it is suppressed in FIXED $O_3$. Furthermore, the November lower stratosphere was dynamically warmed in FIXED $O_3$, which partly offset the radiative cooling effect of ozone depletion.

- Prescribing the CMIP6 ozone field, which is not consistent with the dynamics in FOCI, alters the propagation of planetary waves from the troposphere and changes the location of suppressed wave breaking induced by the ozone hole.

- The ozone-induced austral spring polar cap cooling in the lower stratosphere is weaker in FIXED $O_3$ than in INTERACT $O_3$. The cooling trends estimated from the IGRA radiosonde observations and from the ERA5 reanalysis are stronger than those obtained from the FIXED $O_3$ simulations and fall at the lower end of the range of trends simulated by INTERACT $O_3$.

- The acceleration of the stratospheric jet in response to ozone depletion is also weaker in FIXED $O_3$ than in INTERACT $O_3$ and it agrees better with the estimate from the ERA5 reanalysis. In contrast, the tropospheric jet trend differences between FIXED $O_3$ and INTERACT $O_3$ fall within the range of internal variability.

- The persistence of the SAM is reduced in FIXED $O_3$ due to the dynamically-inconsistent prescribed ozone field, which represents another new result.

The temperature and dynamical response to ozone depletion simulated by FOCI is in good agreement with previous studies. The austral spring lower stratospheric cooling is the signature of the formation of the Antarctic ozone hole, a direct radiative effect also reported by Mahlman et al. (1994); Li et al. (2008); Stolarski et al. (2010); Keeble et al. (2014); Ivy et al. (2016) and Lubis et al. (2016). The November-December warming above the ozone hole represents a dynamical response to ozone loss and is in agreement with the findings of Mahlman et al. (1994); Li et al. (2008); Stolarski et al. (2010); Keeble et al. (2014) and Ivy et al. (2016). This confirms the ability of the model to accurately capture the stratospheric temperature response to ozone depletion. The strong and significant intensification of the polar night jet in austral spring and that of the surface westerlies in austral summer simulated by FOCI as a result of Antarctic ozone loss, combined with the weak and insignificant response of the surface westerlies to the increase in GHG, bring robustness to the conclusions of previous studies (Arblaster and Meehl, 2006; McLandress et al., 2010; Polvani et al., 2011; Previdi and Polvani, 2014; World Meteorological Organization, 2018) that ozone depletion is the main driver of SH stratospheric and tropospheric dynamical changes over the later part of the twentieth century. This is likely to change in the twenty first century, as even higher GHG levels will increase the GHG impact on the atmospheric circulation, while the ban on CFCs emissions following the Montreal protocol is expected to result in a recovery of the ozone hole, driving circulation changes in the opposite direction to those that occurred over the historical period (Perlwitz et al., 2008; Son et al., 2008; McLandress et al., 2010; Eyring et al., 2013).

The strengthening of the westerly winds in response to ozone depletion delayed the breakdown of the polar vortex, which allowed planetary waves to propagate higher into the stratosphere in November and for a longer period of time in December. This resulted in stronger wave dissipation in the upper stratosphere in November and in the middle stratosphere in December, leading to the concomitant intensification of the residual circulation, in agreement with the findings of previous studies that used CCMs (Li et al., 2008; McLandress et al., 2010; Li et al., 2010; Keeble et al., 2014; Oberländer-Hayn et al., 2015; Polvani et al., 2018; Abalos et al., 2019). A new result of this study is that the intensification of the residual circulation is weaker when the CMIP6 ozone field is prescribed and, furthermore, it extends farther down in the lower stratosphere in November. FIXED $O_3$ is not able to reproduce the November lower stratosphere residual circulation weakening found in INTERACT $O_3$ in agreement with other CCMs (Li et al., 2008; McLandress et al., 2010; Lubis et al., 2016). This is an essential aspect in explaining the weaker lower stratospheric cooling in response to ozone depletion in FIXED $O_3$. The enhanced lower stratospheric polar

cap downwelling results in dynamical warming, which reduces the radiative cooling due to ozone loss in November. The consequence is that FIXED $O_3$ exhibits a weaker spring lower stratospheric cooling compared to INTERACT $O_3$.

The performance of simulations prescribing the CMIP6 ozone field was tested here for the first time against simulations using the same model, but in the version that includes interactive ozone chemistry. We found significant differences in the response of the stratospheric temperature and dynamics to ozone depletion in INTERACT $O_3$ and FIXED $O_3$, by comparing the spread of the individual ensemble member trends. Each ensemble member represents one realization of the impacts of ozone depletion using the two methods of imposing ozone changes. Their spread gives an estimate of the range of trends that can be expected from each ensemble, taking into account the internal variability in the model. The caveat of this approach is that we are restricted to only three simulations per ensemble, due to the large computational costs of such simulations. In contrast to the trends in the stratospheric westerly jet, the difference in the tropospheric jet trends between FIXED $O_3$ and INTERACT $O_3$ falls within the internal variability of the model.

Several factors can potentially explain the differences in the ozone-induced stratospheric temperature and circulation trends between INTERACT $O_3$ and FIXED $O_3$. 1) The CMIP6 and FOCI ozone fields exhibit different climatologies, as discussed in Sect. 4.1. Neither the climatological CMIP6 ozone hole, nor its variability are consistent with FOCI's dynamics. The fact that the prescribed ozone hole is displaced in relation to the simulated polar vortex alters the propagation of planetary waves from the troposphere to the stratosphere and therefore leads to changes in the dynamical response to ozone depletion. This results in different dynamical heating rate trends in FIXED $O_3$ and INTERACT $O_3$. 2) The CMIP6 and FOCI ozone fields exhibit different trends. The austral spring polar cap ozone and, consequently, the SW heating rate trends are stronger in FIXED $O_3$ than in INTERACT $O_3$. In contrast, the temperature trends are weaker in FIXED $O_3$ than in INTERACT $O_3$. We therefore conclude that the difference in the imposed ozone trends cannot explain the difference in the temperature trends. 3) The CMIP6 ozone field is interpolated from monthly values to the model time step and the studies of Sassi et al. (2005) and Neely et al. (2014) showed that this can lead to lower temperature trends when monthly ozone fields are prescribed. In this case, the SW heating rate trend would also be weaker in FIXED $O_3$ than in INTERACT $O_3$. However, the FIXED $O_3$ SW heating rate trend is stronger in our study, in line with the stronger ozone trend. Therefore, the monthly resolution of the prescribed CMIP6 ozone field cannot explain the weaker temperature trend in FIXED $O_3$. 4) Feedbacks between ozone, radiation and dynamics cannot occur in FIXED $O_3$. In a recent study with a different model, Haase et al. (2020) used the daily three-dimensional ozone from the interactive chemistry version and prescribed it to the version without interactive chemistry. Despite the fact that the ozone and SW heating rates were the same in their two ensembles, they still found differences between the SH polar cap lower stratospheric temperature and dynamical heating rate trends and attributed these differences to the missing ozone-related feedbacks when ozone is prescribed. These ozone-radiative-dynamical feedbacks are also missing in our FIXED $O_3$ ensemble and might therefore contribute to the differences in the stratospheric temperature and dynamics trends between FIXED $O_3$ and INTERACT $O_3$.

Although the results presented here were obtained using a single model, FOCI, they suggest that prescribing the CMIP6 ozone field leads to weaker simulated effects of ozone depletion in the Southern Hemisphere stratosphere compared to computing the ozone chemistry interactively. This is because the features of the prescribed ozone field are likely to differ in position

and variability from the simulated dynamics, particularly in models that were not involved in the generation of the CMIP6 ozone field. While the effects of prescribing the CMIP6 ozone field on the tropospheric westerly jet cannot be distinguished from internal variability in the current study, they might get more important in the twenty-first century when ozone recovery occurs.

*Data availability.* The model data used in this study can be found at https://zenodo.org/record/3931507. The CMIP6 ozone field prescribed to the simulations that do not calculate the ozone chemistry interactively is publically available at http://blogs.reading.ac.uk/ccmi/forcing-databases-in-support-of-cmip6/. The IGRA radiosonde observations are publically available at https://www1.ncdc.noaa.gov/pub/data/igra/. The ERA5 dataset is available for public use at https://cds.climate.copernicus.eu/#!/search?text=ERA5.

*Author contributions.* II, KM and AB designed the study and the experimental set up. SW, II and JH carried out the model simulations. II
carried out the analysis and all authors discussed the results. II wrote the manuscript with contributions from all co-authors.

*Competing interests.* The authors declare that they have no conflict of interest.

*Acknowledgements.* This study was funded by the German Federal Ministry of Education and Research (Bundesministerium für Bildung und Forschung - BMBF) through the SPACES-II CASISAC project, grant number 03F0796A. The model simulations were performed with resources provided by the North-German Supercomputing Alliance (HLRN). We thank Isla Simpson for her help in the calculation of the
815 Southern Annular Mode timescale.

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

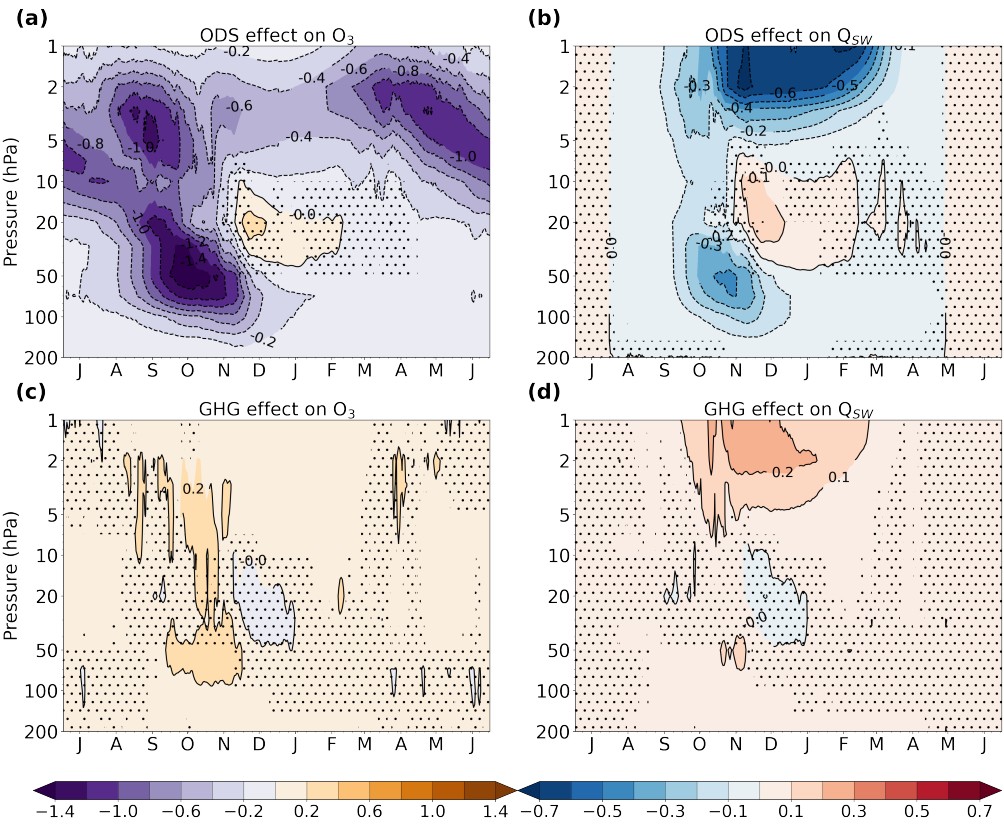

**Figure 1.** Seasonal cycle of the difference between REF and NoODS (a and b) and between REF and NoGHG (c and d) in the ozone volume mixing ratio (a and c, in ppmv) and the SW heating rate (b and d, in K day$^{-1}$) averaged over the polar cap (70°S-90°S) for the period 1978 – 2002. Stippling masks values that are not significant at the 95% confidence interval. The letter corresponding to each month marks the middle of that month.

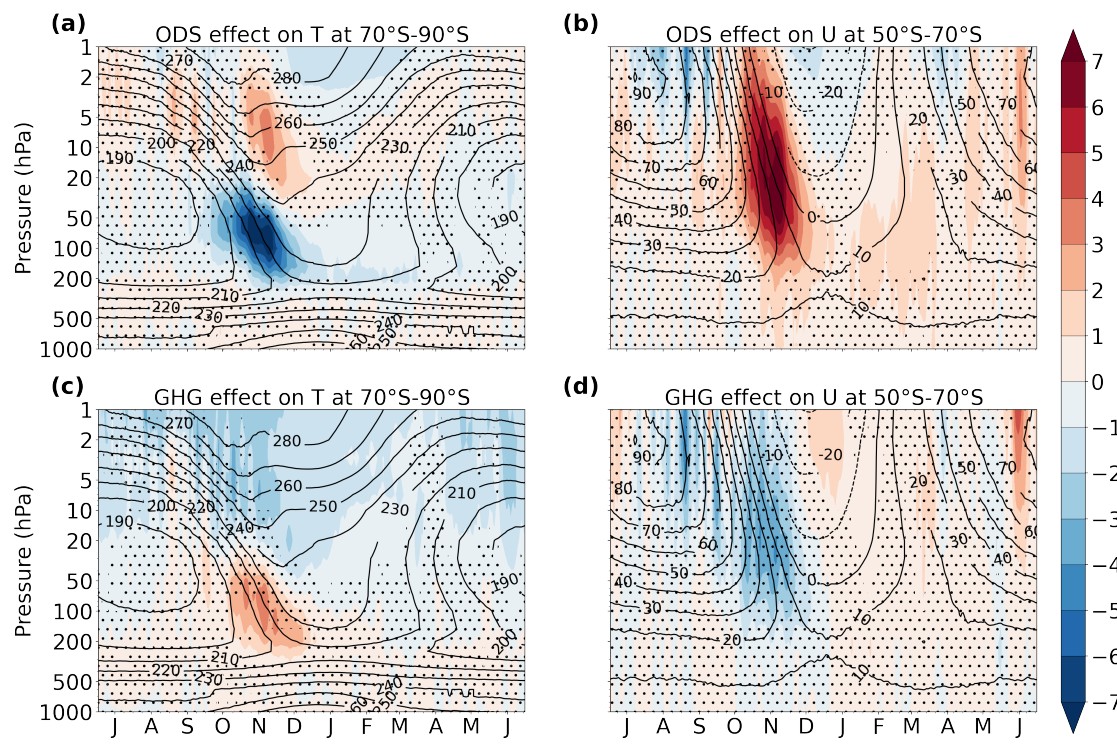

**Figure 2.** Seasonal cycle of the difference between REF and NoODS (a and b) and between REF and NoGHG (c and d) in the polar cap (70°S-90°S) temperature (a and c, in K) and in the mid-latitude (50°S-70°S) zonal wind (b and d, in m s$^{-1}$) for the period 1978 – 2002 (color shading). Stippling masks values that are not significant at the 95% confidence interval. Contours show the corresponding climatological temperature and zonal wind from REF. The letter corresponding to each month marks the middle of that month.

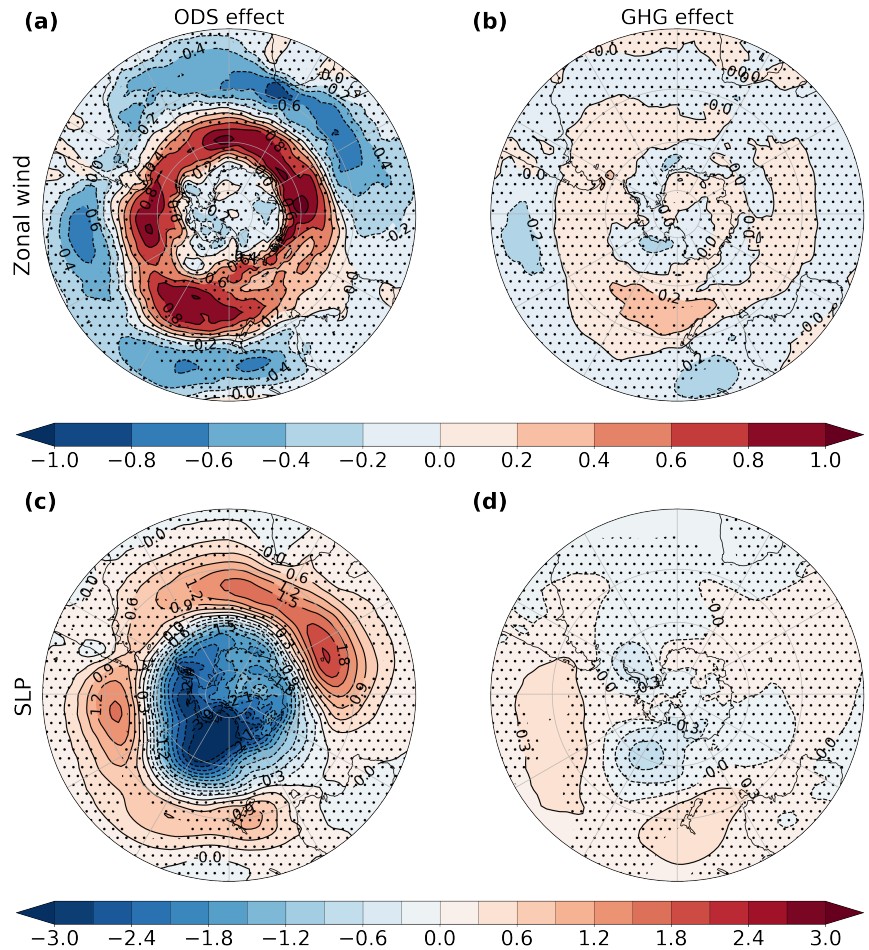

**Figure 3.** Polar stereographic maps of the November-December difference between REF and NoODS (a and c) and the annual mean difference between REF and NoGHG (b and d) in the surface zonal wind (a and b, in m s$^{-1}$) and in sea level pressure (c and d, in hPa) for the period 1978 – 2002. Stippling masks values that are not significant at the 95% confidence interval.

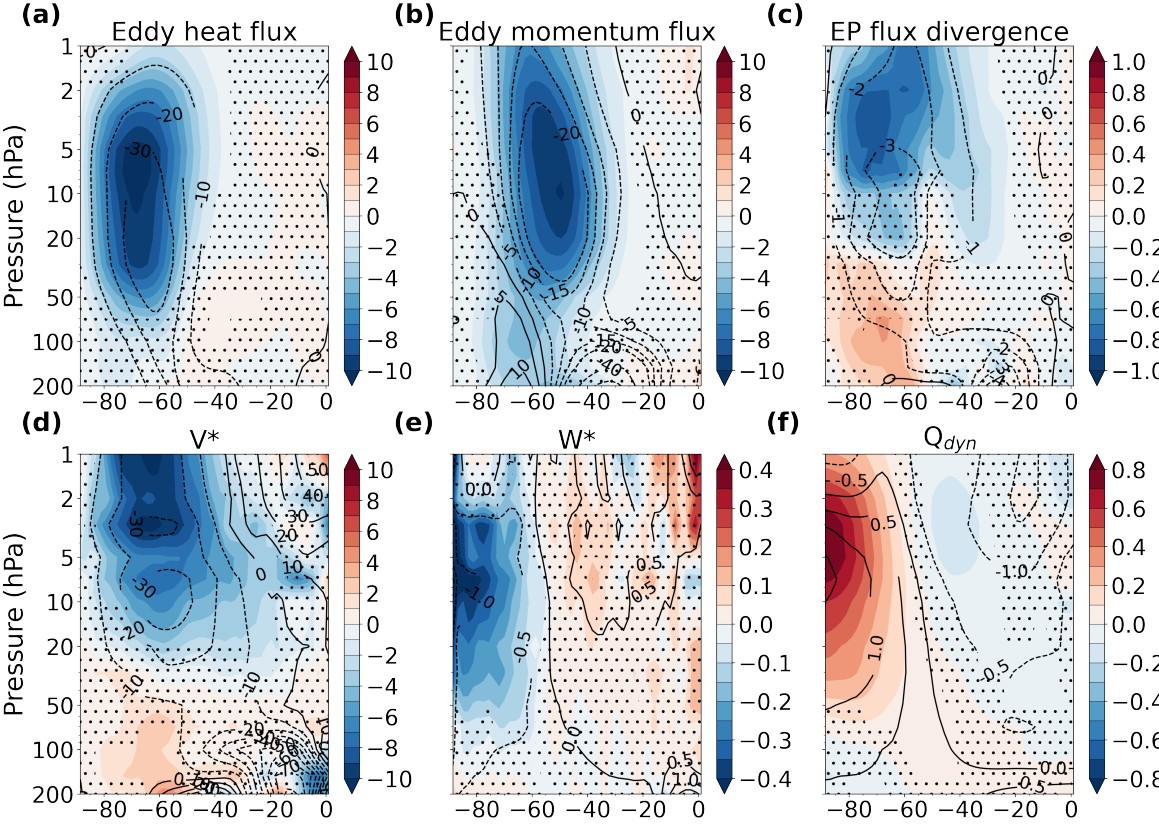

**Figure 4.** Latitude-height November difference between REF and NoODS in the eddy heat flux (a, in K m s$^{-1}$), the eddy momentum flux (b, in m$^2$s$^{-2}$), the divergence of the EP flux (c, in m s$^{-1}$ day$^{-1}$), the meridional residual velocity (d, in cm s$^{-1}$), the vertical residual velocity (e, in mm s$^{-1}$) and in the dynamical heating rate (f, in K day$^{-1}$) for the period 1978 – 2002 (color shading). Contours in each panel show the corresponding climatology from REF. Stippling masks values that are not significant at the 95% confidence interval.

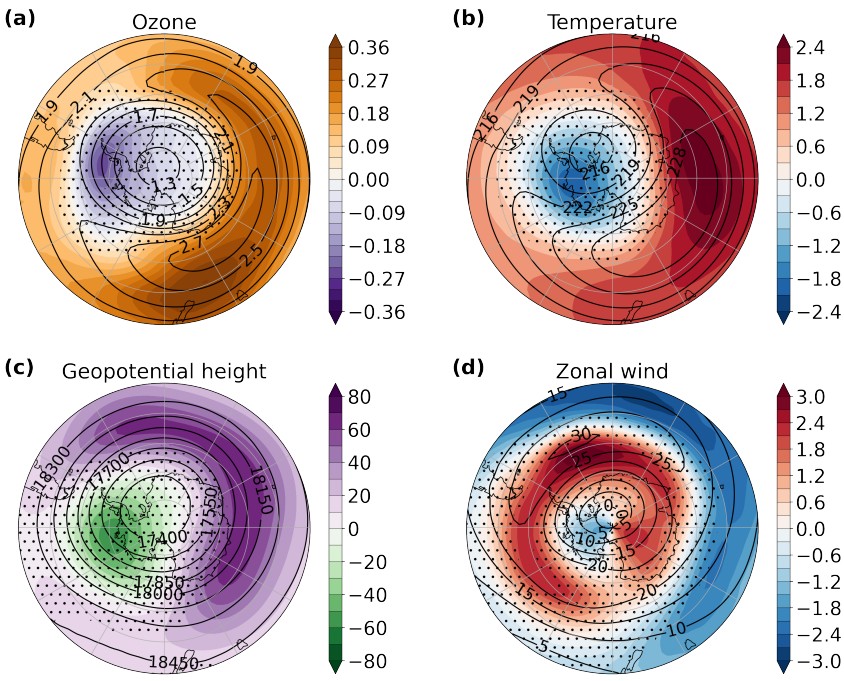

**Figure 5.** Polar stereographic maps of the difference between INTERACT $O_3$ and FIXED $O_3$ in the October mean ozone (a, in ppmv) and the November mean temperature (b, in K), geopotential height (c, in m) and zonal wind (d, in m s$^{-1}$) at 70 hPa (color shading). The stippling masks regions that are not significant at the 95% confidence interval. The overlaying contours mark the 1958-2013 INTERACT $O_3$ climatology of each respective variable and month.

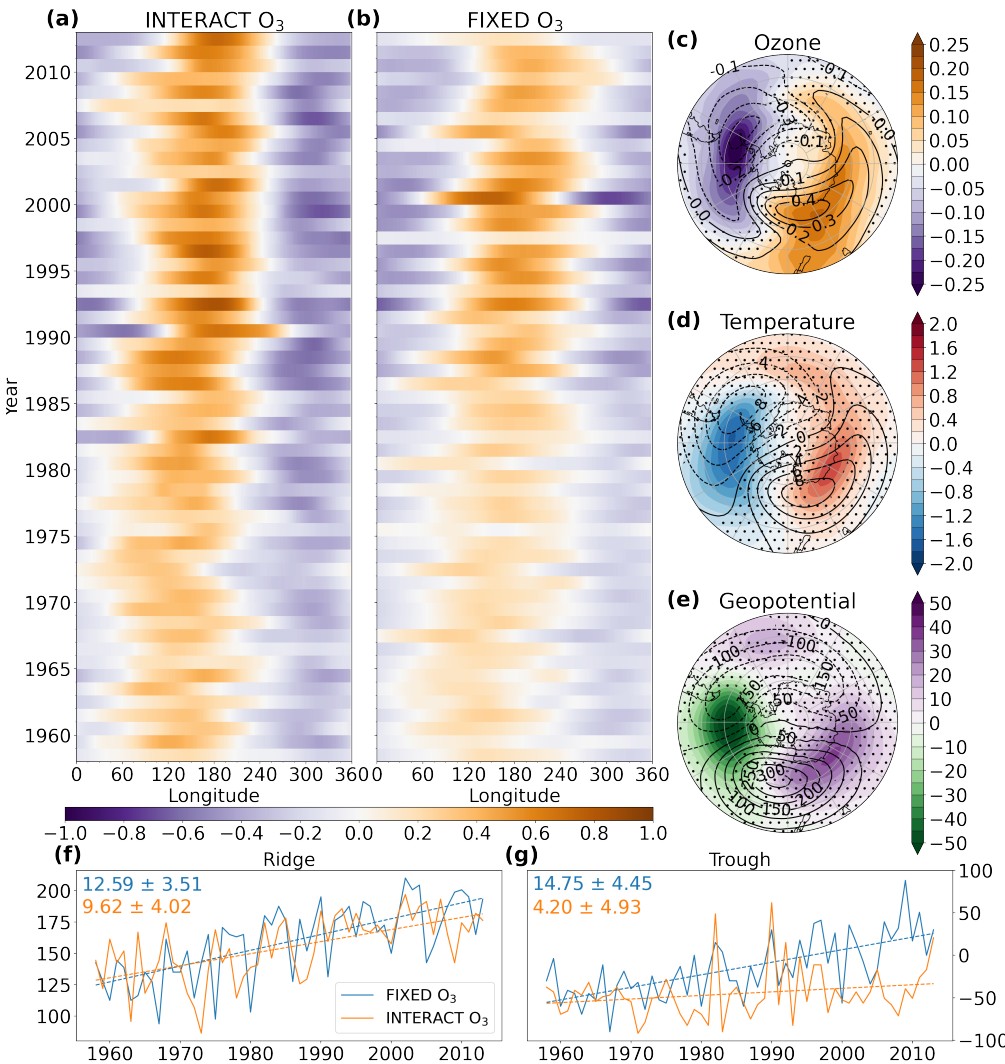

**Figure 6.** Hovmöller diagram of the October anomalies from the zonal mean ozone volume mixing ratio (in ppmv) in INTERACT $O_3$ (a) and FIXED $O_3$ (b) averaged over $60°$S-$70°$S and maps of the October difference between INTERACT $O_3$ and FIXED $O_3$ in the anomalies from the zonal mean ozone volume mixing ratio (c, in ppmv), temperature (d, in K) and geopotential height (e, in m) at 70hPa for the period 1958-2013 (color shading). The stippling in panels c-e masks regions that are not significant at the 95% confidence interval. The overlaying contours mark the INTERACT $O_3$ 1958-2013 average anomalies from the zonal mean for each respective variable. Timeseries of the longitude of the ozone ridge maximum (f) and of the ozone trough minimum (g) for INTERACT $O_3$ (solid orange lines) and FIXED $O_3$ (solid blue lines), together with their corresponding trends for the period 1958-2013 (dashed lines). The values of the trends is in units of $°$ of longitude per decade and their 95% confidence interval according to a two-tailed t-test is given in the upper left corner of each panel.

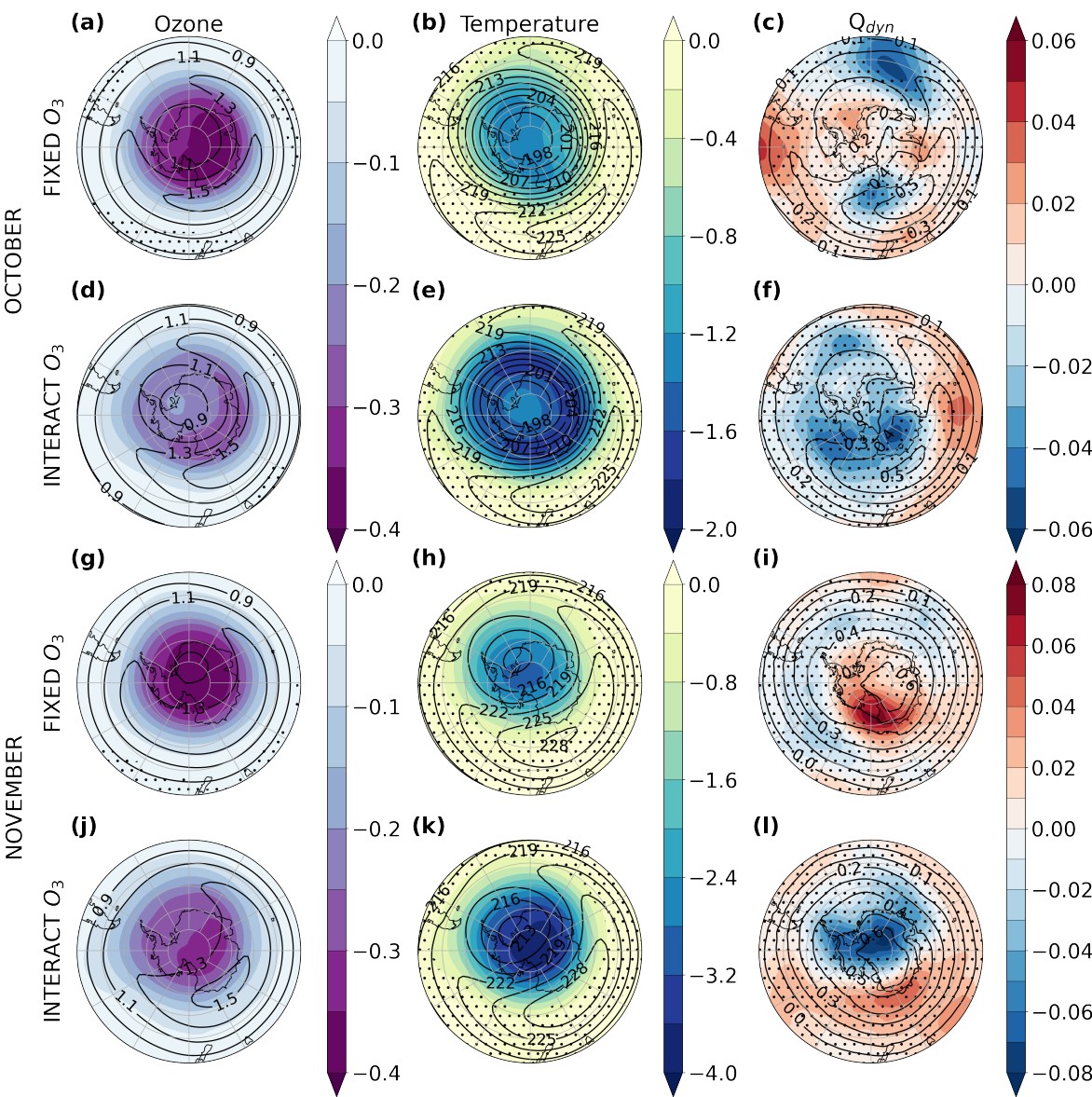

**Figure 7.** Polar stereographic maps of the October (a-f) and November (g-l) trends in ozone (a, d, g, j, in ppmv per decade), temperature (b, e, h, k, in K per decade) and dynamical heating rate (c, f, i, l, in K day$^{-1}$ per decade) at 100 hPa for FIXED O$_3$ (a-c and g-i) and INTERACT O$_3$ (d-f and j-l) over 1958-2002 (color shading). Stippling masks regions where the trends are not significant at the 95% confidence level. The overlaying contours show the respective climatologies for 1958-2002.

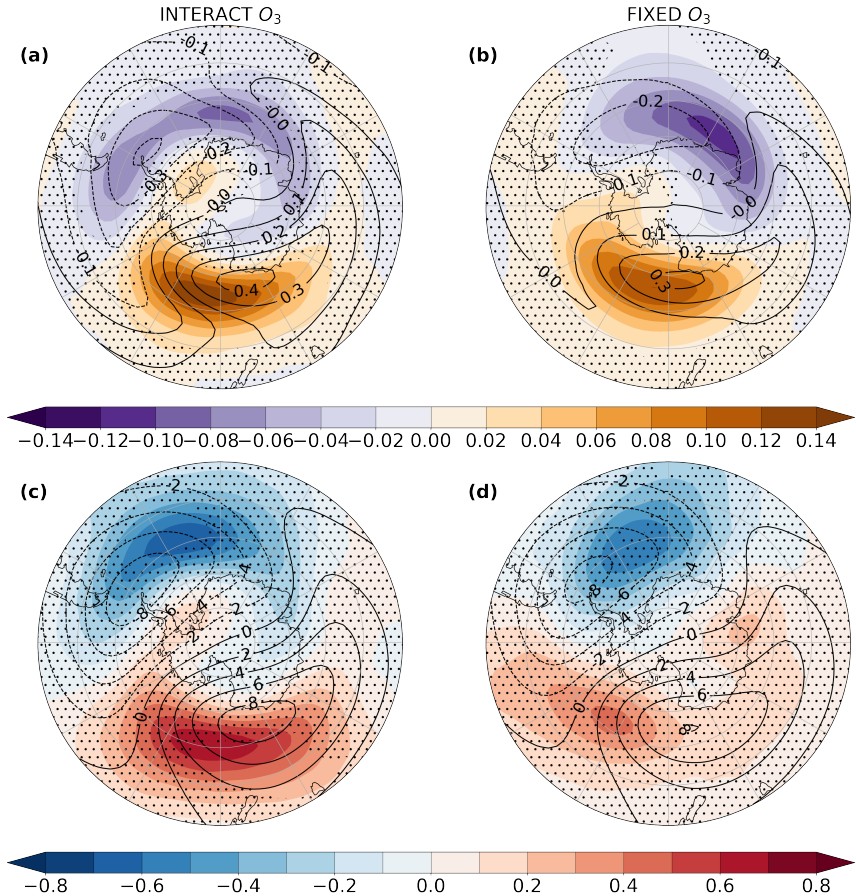

**Figure 8.** Polar stereographic maps of the October 70 hPa trends in INTERACT $O_3$ (a and c) and FIXED $O_3$ (b and d) ozone (a and b, in ppmv per decade) and temperature (c and d, in K per decade) anomalies from the zonal mean for the period 1958-2002 (color shading). Stippling masks regions where the trends are not significant at the 95% confidence level. The overlaying contours show the corresponding October climatologies.

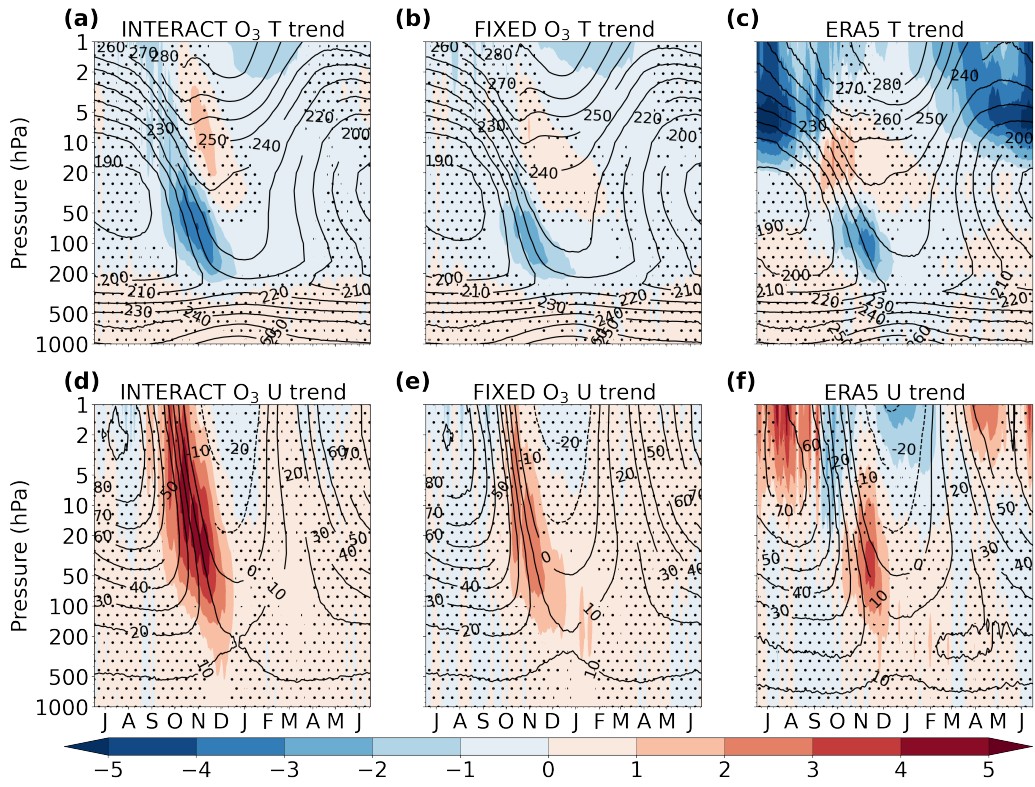

**Figure 9.** Seasonal cycle of the INTERACT $O_3$ (a and d), FIXED $O_3$ (b and e) and ERA5 (c and f) trends in polar cap (65°S-90°S) temperature (a-c, in K per decade) and in the mid-latitude (50°S-70°S) zonal wind (d-f, in m s$^{-1}$ per decade) for the period 1958 – 2002 (color shading). Stippling masks regions where the trends are not significant at the 95% confidence level. The overlaying contours show the corresponding climatological seasonal cycle for 1958-2002. The letter corresponding to each month marks the middle of that month.

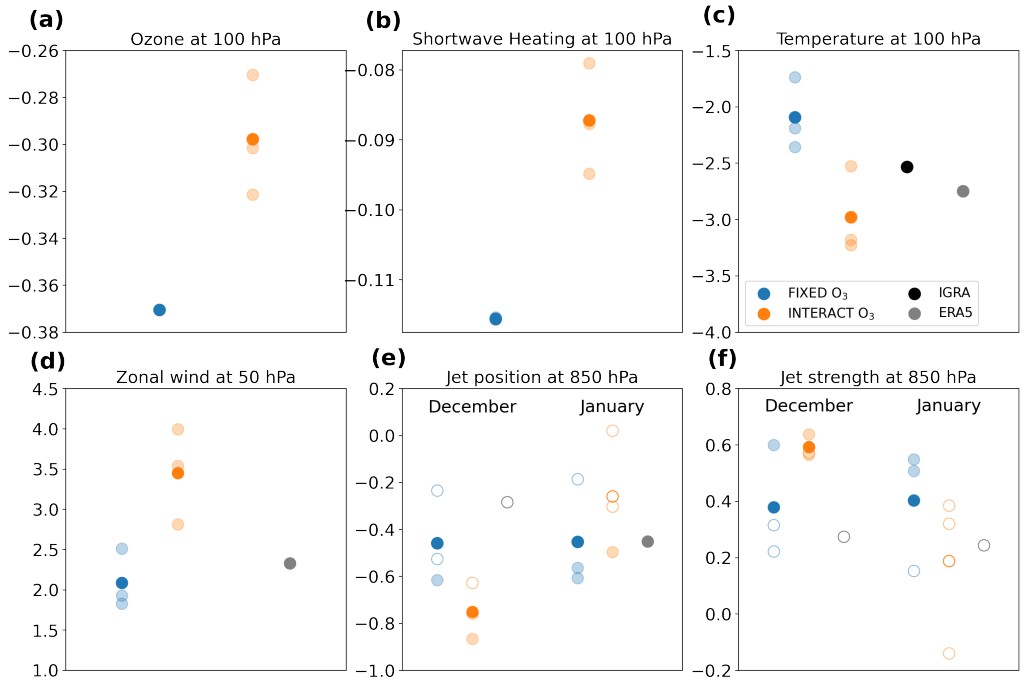

**Figure 10.** Trends in November 100 hPa ozone (a, in ppmv per decade), SW heating rate (b, in K day$^{-1}$ per decade), temperature (c, in K per decade), 50 hPa zonal wind (d, in m s$^{-1}$ per decade) and December and January 850 hPa jet position (e, in ° per decade) and strength (f, in m s$^{-1}$ per decade). The ozone and SW heating rate are averaged over 65°S-90°S, the temperature is averaged over 65°S-80°S, the zonal wind is averaged over 50°S-70°S and the jet's position is given as the latitude where the zonal mean maximum occurs. FIXED O$_3$ trends are shown in blue, INTERACT O$_3$ trends are shown in orange, ERA5 trends are shown in grey and the IGRA temperature trend is shown in black. Ensemble mean trends are depicted by the dark-colored circles, while the individual members' trends are depicted in faded colors. Trends that are significant at the 95% confidence level are marked by filled circles.

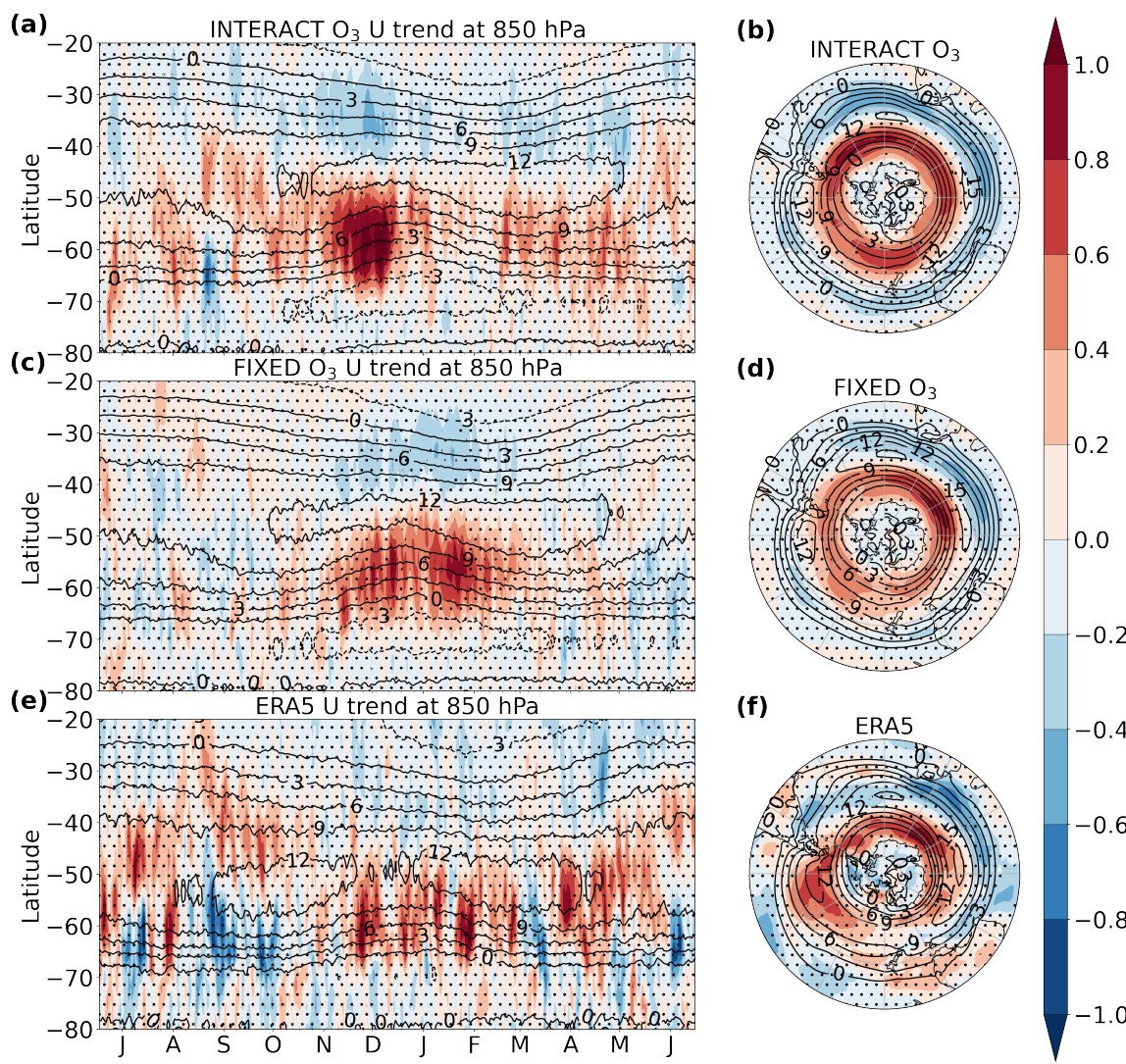

**Figure 11.** Seasonal cycle (a, c and e) and December-January polar stereographic maps (b, d and f) of the 850 hPa zonal wind trend (in m s$^{-1}$ per decade) in INTERACT O$_3$ (a and b), FIXED O$_3$ (c and d) and ERA5 (e and f) for the period 1958-2002 (color shading). Stippling masks regions where the trends are not significant at the 95% confidence level. The overlaying contours show the corresponding climatologies for 1958-2002. The letter corresponding to each month marks the middle of that month.

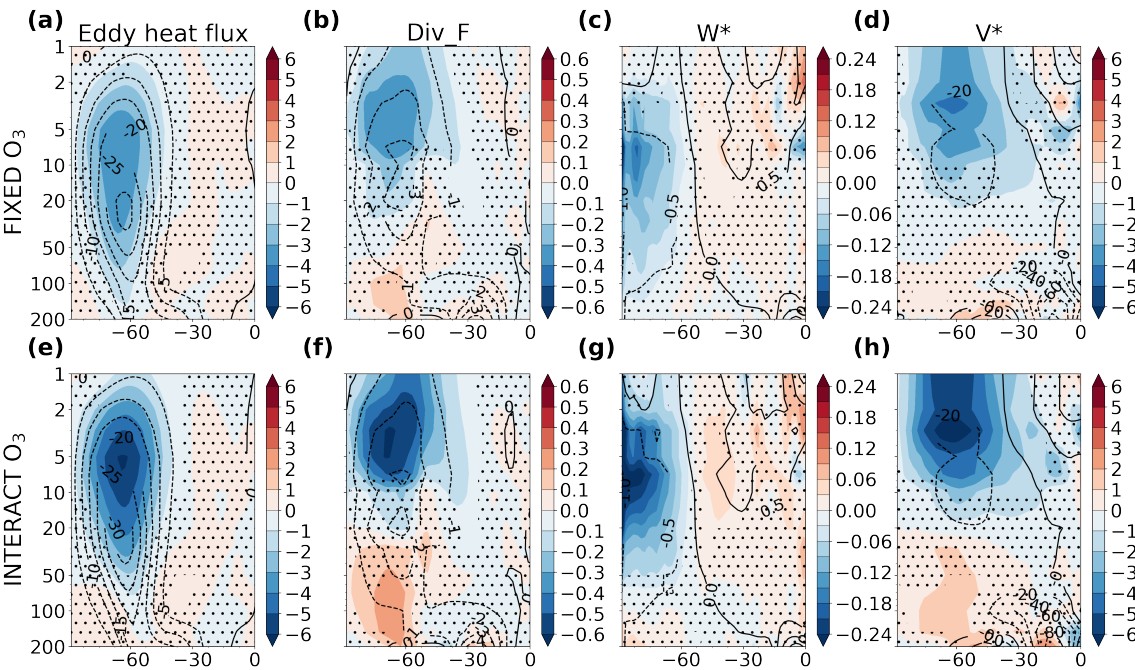

**Figure 12.** Latitude-height November trends in the eddy heat flux (a and e, in K m s$^{-1}$ per decade), the divergence of the EP flux (b and f, in m s$^{-1}$day$^{-1}$ per decade), the vertical residual velocity (c and g, in mm s$^{-1}$ per decade and the meridional residual velocity (d and h, in cm s$^{-1}$ per decade) for the period 1958-2002 in FIXED O$_3$ (a-d) and INTERACT O$_3$ (e-h). The overlaying contours in each panel show the corresponding climatologies. Stippling masks the trends that are not significant at the 95% confidence interval.

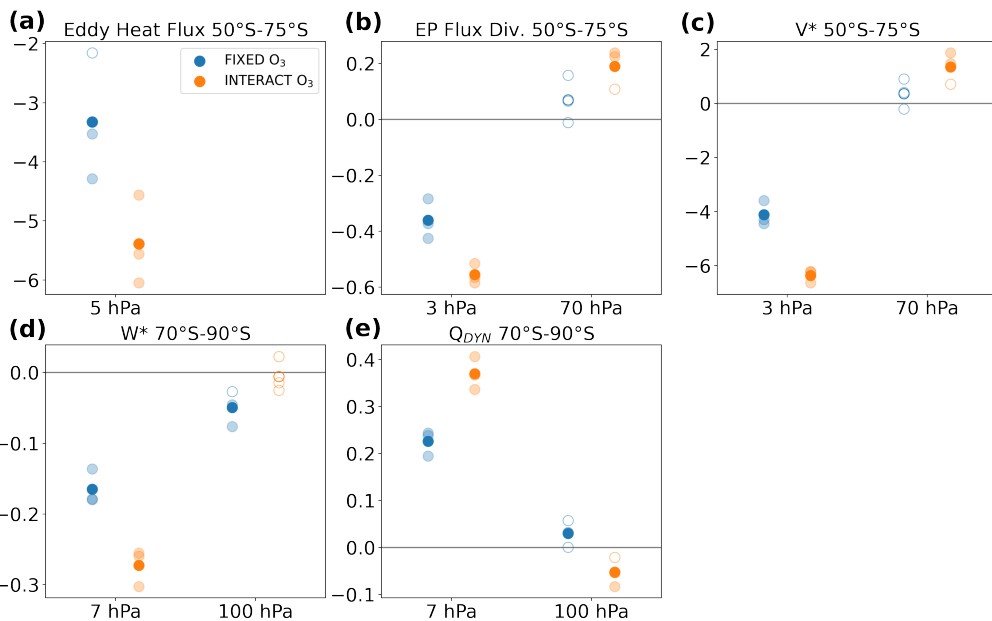

**Figure 13.** Trends in November eddy heat flux (a, in $\mathrm{K\,m\,s^{-1}}$ per decade), EP flux divergence (b, in $\mathrm{m\,s^{-1}day^{-1}}$ per decade), meridional residual velocity (c, in $\mathrm{cm\,s^{-1}}$ per decade), vertical residual velocity (d, in $\mathrm{mm\,s^{-1}}$ per decade) and dynamical heating rate (e, in $\mathrm{K\,day^{-1}}$ per decade). The eddy heat flux, EP flux divergence and meridional residual velocity are averaged over 50°S-75°S and the vertical residual velocity and the dynamical heating rate are averaged over 70°S-90°S. FIXED $O_3$ trends are shown in blue and INTERACT $O_3$ trends are shown in orange. Ensemble mean trends are depicted by the dark-colored circles, while the individual members' trends are depicted in faded colors. Trends that are significant at the 95% confidence level are marked by filled circles.

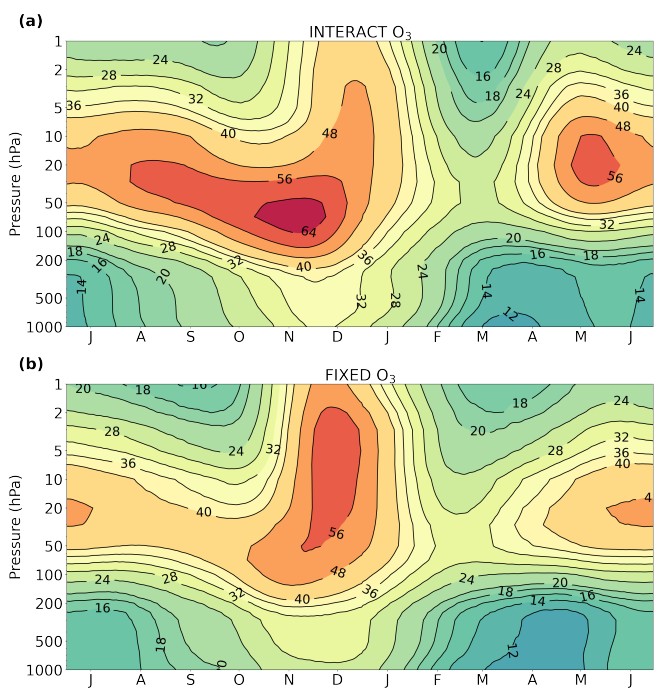

**Figure 14.** Seasonal cycle of the SAM timescale in INTERACT $O_3$ (a) and FIXED $O_3$ (b). Note that the contour interval is non-linear.

**Table 1.** Overview of the FOCI ensembles used in this study. Each ensemble consists of 3 simulations which vary in their initial conditions.

| Ensemble | Ozone chemistry | GHG | ODS | Analysis period |
|---|---|---|---|---|
| FIXED $O_3$ | prescribed CMIP6 | historical CMIP6 | historical CMIP6 | 1958-2002 |
| INTERACT $O_3$ | interactive | historical CMIP6 | historical CMIP6 | 1958-2002 |
| REF | interactive | historical CMIP6 | historical CMIP6 | 1978-2002 |
| NoODS | interactive | historical CMIP6 | fixed at 1960s level | 1978-2002 |
| NoGHG | interactive | fixed at 1960s level | historical CMIP6 | 1978-2002 |