# Peer review of "Effects of prescribed CMIP6 ozone on simulating the Southern Hemisphere atmospheric circulation response to ozone depletion"

_Atmospheric Chemistry and Physics, 2020_

## Referee Comment (RC1) · Anonymous Referee #1 · 2 Nov 2020

Summary: Overall, I find this paper a very enjoyable read. The authors use a new model (FOCI) to address the impact of anthropogenic drivers (increasing GHGs, ozone depletion) on SH climate, and specifically to characterize the role of interactive ozone when interactive chemistry is the only point of difference between two model constellations. The authors have done a good job communicating their story of significant differences in stratospheric and tropospheric dynamics depending on how ozone is handled. I think their methodology is sound and the results are plausible and in large parts backed up by existing literature (although few other works have presented these results with such clarity).

[Figure]

Apart from a few minor issues detailed below, my only major point of criticism is that the authors should have used the simulations already presented by Haase et al. (2020) as a further line of evidence. In these simulations (only mentioned at the end of the manuscript) the ozone field is taken from a CHEM-ON simulation of FOCI and used at daily resolution to force an offline simulation with FOCI. Using these simulations would address a question that I had reading the manuscript, that various dynamical differences between the CHEM-ON and CHEM-OFF simulations might not have been caused directly by the method of treatment of ozone but rather by possibly substantial differences in the background ozone climatology. Such differences would be minimized in the above comparison; only differences to do with mismatches between the state of ozone and the state of the polar vortex would remain. (A better comparison still might be to take the ozone field from FOCI, filter out interannual variations and use it at monthly resolution, to prescribe ozone in as similar a way as it gets to the CMIP6 climatology, but with systematic ozone differences removed. That would make both ensembles of simulations comparable to the majority of CMIP6 historical simulations that have used the CMIP6 ozone climatology.)

Essentially, adding this simulation ensemble would allow the authors to decompose any differences in trends into contributions due to the background ozone climatology and due to consistency (or not) between ozone and dynamics, which are two quite different explanations that the authors cannot really distinguish between in the paper as it stands. Since the simulation(s) needed for this already exist, I feel this is not an enormously large request to make (although it might make the text longer and the figures more complex).

A further, less fundamental question relates to the treatment of radiatively active gases other than ozone in CHEM-OFF. How do you treat water vapour, methane, and nitrous oxide when chemistry is turned off, in such a way as to minimize differences between the CHEM-OFF and CHEM-ON simulations? Or is chemistry running in all model variants and just a different flavour of ozone is fed into radiation?

[Figure]

And finally, you only mention $CO_2$ and methane as 'GHGs' in the set-up of the 'No-ODS' simulations. How about $N_2O$? Is that considered an ODS? Technically it is, but it affects ozone very differently from the halogenated ODSs (see e.g. https://doi.org/10.5194/acp-18-1091-2018). Typically ODSs are taken to be chlorinated and brominated halocarbons controlled by the Montreal Protocol. Please clarify.

Final question: Are you planning to contribute this model to CMIP6, given the large effort put into producing the PI spin-up and the historical simulations?

Minor comments: P2L29: "at destroying ozone"

P2L52: I think it's controversial whether East Antarctica actually experienced cooling, considering the difficulties with measuring temperature there (distinguishing cloud from ice in IR-measurements, sparsity of ground-based measurements). For a "grey-literature" comment on this see http://www.realclimate.org/index.php/archives/2004/12/antarctic-cooling-global-warming/. It's more robust to assert that the rate of warming in Antarctica exhibits large regional variations and that a large warming of much of West Antarctica has occurred.

P3L75: I find "zonally oriented asymmetries" confusing. I suggest dropping this phrase and just call them "zonal asymmetries".

P4L127: Worth noting that these were not just "different climate models" but actually two different chemistry-climate models whose results for "historical" ozone were averaged to form the CMIP6 ozone forcing dataset. You want to cite https://doi.org/10.1002/2017GL076770 for this.

P6L1: I infer from "$CO_2$ emissions" that this model is run using an interactive carbon cycle? Please state in the text if that is the case. Using methane "emissions" is also unusual in this context; typically methane VMRs are prescribed at the surface. "Emissions" are typically only used for tropospheric ozone and aerosol precursors. Also, please state explicitly how $N_2O$, HFCs, PFCs and other minor greenhouse gases are

treated (see above).

P10L298: "indicates"

P13L407ff: I can see westward shifts in CHEM-ON and CHEM-OFF (figure 6), but the difference in the rates of progression is not obviously discernible to me. Could you perhaps think of a way of visualizing this better, and perhaps formalize that the trends are significantly different? You may want to cite https://doi.org/10.5194/acp-17-14075-2017 here.

P16L507: "shifted"

Figure 4 and elsewhere: My inclination is to avoid introducing scaling factors into colour bars (e.g. $10^{-2}$ m s$^{-1}$, $10^{-4}$ m s$^{-1}$) and just state unscaled units (cm/s, mm/s). This is also done elsewhere but does not improve clarity, I find. Also figure 5, panel a. Why not have the scale range from $-0.36$ to $+0.36$?

---

## Author Comment (AC1) · 9 Nov 2020

Dear Referee #1,

We thank you for your comments and suggestions. We will address all your comments in the response to all reviewers once the interactive discussion is completed. At this time, we would like to point out that there has been a misunderstanding regarding the simulations of Haase et al. (2020). The simulations they use, in which the ozone field from an interactive chemistry simulation is prescribed to a simulation in which the interactive chemistry is turned off are performed with the CESM1-WACCM model, not with FOCI. Therefore, these simulations are not directly comparable to the ones presented

in our manuscript and we would have to perform an ensemble of three additional simulations in order to assess the differences in dynamics when the background ozone climatology stays the same in Chem ON and Chem OFF. Performing the additional three simulations is not feasible at this time, as they require a large amount of computing time and storage space, which needs to be planned for in advance. Therefore, we unfortunately cannot extend our analysis in the way you suggested.

Best regards,

Ioana Ivanciu
* * *

---

## Referee Comment (RC2) · Anonymous Referee #3 · 30 Nov 2020

The paper reports on the impact of ozone depletion and greenhouse gases on atmospheric circulation trends in the Southern Hemisphere in the new FOCI model. In addition, the paper also documents the impact of prescribing (rather than simulating) an ozone field on the effects of ozone depletion, by comparing ensembles with interactive ozone against ensembles using the CMIP6 ozone forcing. The authors conclude that FOCI captures the effects of ozone depletion and GHGs on the circulation. In addition, they also conclude that prescribing ozone rather than simulating one interactively leads to a weaker tropospheric response to ozone depletion. Based on these results, the paper claims that climate models prescribing CMIP6 ozone will underestimate the historical ozone-induced dynamical changes in the Southern Hemisphere.

The subject of the paper is of relevance and interest for the readership at Atmospheric Chemistry and Physics. The paper is well written and the analysis is detailed and nicely discussed. However, some of the evidence provided is not convincing and as a result, some of the implications of this paper (models without interactive chemistry underestimating the effects of ozone depletion) are over-stated, especially concerning the limitations of imposing a prescribed ozone in historical simulations. The authors need to provide more convincing evidence to support some of the claims, or substantially revise some of them (perhaps tone them down!). Hence, I recommend major revisions, as detailed below.

MAJOR ISSUES:

1) This paper does not really provide a clean isolation of the 'ozone feedback'. Ozone from CMIP6 is substantially different from the ozone simulated by FOCI, as shown in Fig.7. This leads to a systematic bias in the 'CHEM OFF' experiments, as discussed in section 4.1. In addition to differences in climatological values, trends in the prescribed and interactive ozone are also different. Hence, any effects on the variability/trends are a result of these differences, rather than a missing 'ozone-radiative-chemical feedback' in the CHEM OFF ensemble (e.g., L16, L607). If the authors wish to quantify the ozone-chemical feedback, they should compare ensembles using the interactive ozone vs ensembles imposing a (time-varying) ozone, derived from the same model system (FOCI), rather than from other models (CMIP6).

2) The comparison with observations is missing. The paper claims in several instances that the FOCI model 'reliably' captures the effects of ozone depletion (e.g., in the Abstract on L17) and that in simulations with interactive chemistry, the effects of ozone depletion are stronger and closer to the observations (e.g. in the Conclusions near L610). However, this comparison with observations is lacking, as no single observational data-set is shown in the paper, using the same analysis period & statistical method. The authors should directly compare their ensembles (especially REF and CHEM ON) against observations, at least for some of the large-scale circulation metrics, such as zonal mean zonal wind, temperature, near-surface wind (850 hPa), to build confidence in some of their claims regarding the model's skills in capturing observed trends and regarding the increased 'realism' in the simulations with interactive chemistry. I suggest using ERA5 or some other high-quality re-analysis product for this purpose.

3) Inappropriate time-period chosen to analyze the impact of ozone depletion. The authors use the 1958-2013 period to calculate trends in their historical simulations, and they derive the impact of the ODS by taking differences between noODS and REF ensembles (REF is presumably the same as the CHEM ON...??). ODS emissions were phased out in the mid 90s and as a result, ozone depletion trends stopped near the year 2000. Since the beginning of the 21st century, we have already seen the emergence of 'healing' in the ozone layer (Solomon et al., 2016). Recently, it has also been shown that this resulted in a change in the tropospheric circulation trends in the SH (Banerjee et al., 2020). Hence, the trends calculated in this paper do not properly isolate the effects of ozone depletion, as the trends before and after 2000 are probably very different. I would strongly recommend choosing an earlier end-date in the analysis of trends (e.g. early 2000s).

4) No convincing statistics. Aside from direct comparison with observations, we need to make sure differences are really robust across ensemble members. Several studies (e.g., Seviour et al., 2017) have shown how large the variability in the SH can be, and how it can explain differences across transient simulations. Can we make sure the CHEM ON vs OFF differences are really robust in light of this large variability? I would strongly recommend comparing the response to ozone depletion (and most importantly, the CHEM ON vs OFF differences) against the inter-ensemble spread. Ideally, the authors should show the individual members of all ensembles against observations, to give more confidence in the two key statements made by this paper concerning (1) this model reliably reproducing observations in terms of the effects of ozone depletion and (2) its trends being significantly weaker with prescribed (CHEM

[Figure]

OFF) than interactive (CHEM ON) ozone.

5) Lacking discussion of relevant literature. The authors do not sufficiently discuss some key studies, which already looked at role of interactive chemistry in simulating the impact of ozone depletion. One of them was Eyring et al., 2013, which directly compared CMIP5 simulations from CHEM and NOCHEM models. More recently, Seviour et al. (2017) and S. Woo-Son et al. (2018) also extensively analyzed multi-model comparisons (CCMI and CMIP5) in terms of their simulated ozone depletion, and found barely any robust difference between models with and without interactive ozone. These papers should be properly cited and discussed, to provide a more critical and balanced discussion throughout the paper.

6) Unjustified claims regarding underestimation of the effects of ozone depletion in models without interactive chemistry (e.g. see L659). Several papers have shown that actually, imposing or simulating the ozone hole does not make a lot of difference. See for example S.-Woo-Son et al., 2018, as well as Eyring et al., 2013. While it's true that there inter-comparison studies do not cleanly isolate the impact of interactive chemistry alone, they do not see any systematic difference between both class of models (CHEM or NOCHEM), and they span over a wider range of uncertainty, since they look at many different models rather than a single model, as done in this paper. Hence, the implications of this study may be smaller than stated in the paper (e.g. the statement in L21-23 in the Abstract). Moreover, this paper does not properly compare any of the FOCI trends with observations (major comment 2), nor cleanly isolates the importance of interactive chemistry (major comment 1). Hence, the claims about CMIP6 models underestimating the historical ozone-induced changes in the SH are unjustified.

SPECIFIC ISSUES:

L16 "missing ozone radiative dynamical feedbacks" - see major comment 1: The CHEM ON vs OFF comparison rather quantifies the impact of a systematic bias, rather than a true feedback (which could only be quantified by comparing another CHEM OFF

ensemble driven with the ozone forcing from CHEM ON).

L21-23 In light of the discussion given above (points 1, 5-6), I frankly do not find this statement very convincing.

L65 Oehrlein et al. (2020) is another recent relevant paper that studied this problem, as they compare CHEM ON vs OFF experiments strictly having the same ozone climatology. I suggest adding this paper to the reference list.

L83-85 Oehrlein et al., 2020 also explored this. They show that in time-slice simulations with constant forcing, the effect of interactive chemistry on SSWs frequency is not statistically significant. Adding this paper could help making the point about the lack of robustness across different studies on this.

L93 there were also papers showing the contrary, i.e. that models with/without interactive chemistry were very similar in their simulated trends. One paper showing this was Seok-Woo Son et al., 2018. This paper should be cited and discussed.

L114-118 Oehrlein et al., 2020 also studied this problem, using time-slice rather than transient simulations, partly confirming some of the results of Haase and Matthes (2019) but also refuting some others (e.g. the influence on SSW frequency), so I recommend citing this paper here, too.

L125-140 Another problem which is not discussed at all is the vertical interpolation. Interpolating the ozone forcing from CMIP6 which is provided on pressure levels on FOCI's own z-levels may create errors, which can be non-negligible near the tropopause. This can create problems with radiative transfer, as e.g. discussed in Hardimann et al., 2019. Have the authors tested whether this happens, too?

L139 "correctly simulate the effects of ozone depletion" - the authors do not show any observations in this paper. Hence, we cannot really determine whether the REF ensemble (which is the same as CHEM ON I guess?) is really close to observations and whether CHEM OFF is systematically off. I strongly suggest adding one such

analysis. This could be, e.g. by adding trends in jet-latitude or SAM trends and show individual ensembles vs observations, as done e.g. in Seviour et al., 2017.

L160-180 if FOCI by default uses interactive chemistry in the REF experiment, then what is the difference between this ensemble and the CHEM ON ensemble?

L210-220 Is these multiple filtering really needed? Are the results for the SAM sensitive to the way the data is filtered? It would be nice if the authors could comment on this.

L231 see major comment (3) concerning the time-period. The authors should explore the sensitivity of these results to the end year chosen, and a shorter period (e.g., 1958-2000) would probably be more appropriate to explore the effects of ozone depletion.

L256 Actually, this problem has been studies in multiple papers, which looked at the upper stratospheric ozone response to large $CO_2$ forcings in detail; e.g. Haigh and Pyle (1982), Jonsson et al. (2004), Chiodo et al. (2018). I suggest adding these papers here.

L262 This effect (GHGs –> polar cap ozone increase due to faster BDC) has been widely studied in the context of climate sensitivity experiments, imposing large $CO_2$ forcings (2x and 4x$CO_2$), such as e.g. Dietmuller et al. (2014); Nowack et al. (2015); Chiodo et al. (2018). Such results also apply here, although the GHG forcing studied in this paper is much smaller.

L287 "realistically capture" - I recommend adding one figure for the observations, so that the reader can appreciate how close REF (or CHEM ON) are to the observational trend. The validation paper for FOCI (Matthes et al., 2020) did not really show trends in the SH circulation, so this should be done in this paper, since the main message is that FOCI is "able to capture" the impacts of ozone depletion.

L366 "adequately simulates" - same comment as on L287

L375 I would strongly recommend changing the labeling (CHEM OFF) to something more descriptive of what is really used here (CMIP6 O3 forcing). How about CHEM

[Figure]

ON vs FIXO3 CMIP6?

L474 "agree better with observed trends" –> this has not really been shown here, so I am not convinced about this statement. To lend confidence on the results and statements like this throughout the paper, the authors would need to show (1) that all three ensemble members are closer to the observations than CHEM OFF and (2) that they are all significantly different from any of the members in CHEM OFF. This also applies to L610 in the conclusions section.

L494 "the feedbacks between ozone..." As stated in my major comment (1), this simulations set-up does not really cleanly isolate the feedback, as the CMIP6 ozone forcing leads to a systematically different basic state. How can we be sure that these differences are rather due to "biases" introduced by the CMIP6 ozone forcing, rather than a true "feedback"?

L511 this is a very far fetched statement, given that some studies in the past have already shown that CMIP5 models with interactive chemistry (CHEM) do not significantly differ from those that impose the historical ozone forcing in terms of the tropospheric trends (e.g. Eyring et al., 2013; Seviour et al., 2017; Seak-Woo-Son et al., 2018... just to name a few!). To show that CMIP6 is different in this sense, the authors would need to use a different set-up and/or use more models. Otherwise, this is an over-statement which is not justified by the evidence provided in this paper. This also applies to L660-662.

L552-556 see major comment on L511, and major comment (1). This also applies to the statements on L606-607.

REFERENCES

Banerjee et al., 2020; A pause in Southern Hemisphere circulation trends due to the Montreal Protocol, https://doi.org/10.1038/s41586-020-2120-4

Chiodo et al., 2018; The response of the ozone layer to quadrupled $CO_2$ concentra-

tions, DOI:10.1175/JCLI-D-17-0492.1

Dietmuller et al., 2014; Interactive ozone induces a negative feedback in CO2 driven climate change simulations, DOI:10.1002/2013JD020575

Haigh and Pyle, 1982; Ozone perturbation in a two dimensional circulation model, DOI: 10.1002/qj.49710845705

Hardimann et al., 2019; The Impact of Prescribed Ozone in Climate Projections Run With HadGEM3 GC3.1, DOI:10.1029/2019MS001714

Jonsson et al., 2004; Doubled CO2 induced cooling in the middle atmosphere: Photo-chemical analysis of the ozone radiative feedback, DOI:10.1029/2004JD005093

Nowack et al., 2015; A large ozone-circulation feedback and its implications for global warming assessments, DOI:10.1038/nclimate2451

Oehrlein et al., 2020; The effect of interactive ozone chemistry on weak and strong stratospheric polar vortex events, DOI:10.5194/acp-20-10531-2020

Seok-Woo Son et al., 2018; Tropospheric jet response to Antarctic ozone depletion: An update with Chemistry-Climate Model Initiative (CCMI) models, DOI:10.1088/1748-9326/aabf21

Seviour et al., 2017; Robustness of the Simulated Tropospheric Response to Ozone Depletion, DOI:10.1175/JCLI-D-16-0817.1

Solomon et al., 2016; Emergence of healing in the Antarctic ozone layer, DOI: 10.1126/science.aae0061

---

## Author Comment (AC2) · 31 Jan 2021

*Dear Dr. Dameris, dear reviewers,*

*We thank you for your time and for your valuable recommendations that substantially helped us to improve our manuscript. We first summarize the major changes to the manuscript, before providing our point-by-point answers to the comments of the reviewers in blue and italics.*

*As both reviewers pointed out that the set-up of our simulations cannot isolate the role of feedbacks between ozone, radiation and dynamics in the reported differences between simulations with interactive ozone (INTERACT $O_3$) and simulations with prescribed CMIP6 ozone (FIXED $O_3$), we wish to emphasize that this is not the aim of our study. Instead, we aim to investigate how the same model simulates the Southern Hemisphere (SH) effects of ozone depletion, when the ozone field is calculated interactively versus when the CMIP6 ozone field is prescribed, motivated by the fact that many of the models participating in CMIP6 prescribe this ozone field in the absence of an interactive chemistry scheme. We acknowledge that the ozone feedbacks cannot be isolated by our study and that they are just one of several reasons for the different trends in INTERACT $O_3$ and FIXED $O_3$ and we rephrased our statements throughout the manuscript accordingly.*

*Major changes to the manuscript include:*
- *the addition of trend estimates for temperature and zonal winds from the IGRA radiosonde observations and from the ERA5 reanalysis;*
- *two new figures (Figs. 10 and 13) that show the spread of the trends between the individual members of INTERACT $O_3$ and FIXED $O_3$, as well as more figures in the supplement that serve the same purpose. The old Figs. 10 and 13 have been moved to the supplement in order to accommodate the addition of the more important new figures;*
- *a new period over which we compute the INTERACT $O_3$ and FIXED $O_3$ trends, 1958-2002 as opposed to 1958-2013 and a new period over which we computed the differences between REF and NoODS/NoGHG, 1978-2002, as recommended by referee # 3 and motivated below;*
- *the renaming of the Chem ON and Chem OFF ensembles to INTERACT $O_3$ and FIXED $O_3$ as recommended by referee #3.*

*Detailed responses to all the comments of the reviewers are provided in the following. Line numbers refer to the version of the manuscript with tracked changes found at the end of this document.*

**Anonymous Referee #1**

Summary: Overall, I find this paper a very enjoyable read. The authors use a new model (FOCI) to address the impact of anthropogenic drivers (increasing GHGs, ozone depletion) on SH climate, and specifically to characterize the role of interactive ozone when interactive chemistry is the only point of difference between two model constellations. The authors have done a good job communicating their story of significant differences in stratospheric and tropospheric dynamics depending on how ozone is handled. I think their methodology is sound and the results are plausible and in large parts backed up by existing literature (although few other works have presented these results with such clarity).

Apart from a few minor issues detailed below, my only major point of criticism is that the authors should have used the simulations already presented by Haase et al. (2020) as a further line of evidence. In these simulations (only mentioned at the end of the manuscript) the ozone field is

taken from a CHEM-ON simulation of FOCI and used at daily resolution to force an offline simulation with FOCI. Using these simulations would address a question that I had reading the manuscript, that various dynamical differences between the CHEM-ON and CHEM-OFF simulations might not have been caused directly by the method of treatment of ozone but rather by possibly substantial differences in the background ozone climatology. Such differences would be minimized in the above comparison; only differences to do with mismatches between the state of ozone and the state of the polar vortex would remain. (A better comparison still might be to take the ozone field from FOCI, filter out interannual variations and use it at monthly resolution, to prescribe ozone in as similar a way as it gets to the CMIP6 climatology, but with systematic ozone differences removed. That would make both ensembles of simulations comparable to the majority of CMIP6 historical simulations that have used the CMIP6 ozone climatology.)

Essentially, adding this simulation ensemble would allow the authors to decompose any differences in trends into contributions due to the background ozone climatology and due to consistency (or not) between ozone and dynamics, which are two quite different explanations that the authors cannot really distinguish between in the paper as it stands. Since the simulation(s) needed for this already exist, I feel this is not an enormously large request to make (although it might make the text longer and the figures more complex).

*We acknowledge that in this study we cannot distinguish whether the differences in trends are due to ozone related feedbacks or due to the different ozone fields imposed in the INTERACT $O_3$ (former Chem ON) and FIXED $O_3$ (former CHEM OFF) ensembles. However, as we point out in the direct response to your comment, the simulations used in the study of Haase et al. (2020) are performed with a different model, CESM1-WACCM, and not with FOCI. Therefore, we cannot directly use them in our study. Performing an additional three simulations with FOCI following the procedure of Haase et al. (2020) is not feasible at this point, as they require a large amount of computing time and storage space, which needs to be planned for in advance. We have instead adjusted our phrasing throughout the manuscript to reflect the fact that ozone feedbacks are just part of the explanation regarding the different trends in INTERACT $O_3$ and FIXED $O_3$ and that the difference in climatology of the imposed ozone fields is also important. Please also refer to the direct response to your review and to the response to the major comment 1 by referee #3.*

*In the updated manuscript, we now discuss Haase et al. (2020) in more detail in the introduction and in the conclusion.*

*"We attribute this weaker response primarily to a prescribed ozone hole which is different to the model dynamics and is not collocated with the simulated polar vortex, altering the strength and position of the planetary wavenumber one." (lines 18-21)*

*"Several factors can potentially explain the differences in the ozone-induced stratospheric temperature and circulation trends between INTERACT $O_3$ and FIXED $O_3$. 1) The CMIP6 and FOCI ozone fields exhibit different climatologies, as discussed in Sect. 4.1. Neither the climatological CMIP6 ozone hole, nor its variability are consistent with FOCI's dynamics. The fact that the prescribed ozone hole is displaced in relation to the simulated polar vortex alters the propagation of planetary waves from the troposphere to the stratosphere and therefore leads to changes in the dynamical response to ozone depletion. This results in different dynamical heating rate trends in FIXED $O_3$ and INTERACT $O_3$. 2) The CMIP6 and FOCI ozone fields exhibit different trends. The austral spring polar cap ozone and, consequently, the SW heating rate trends are stronger in FIXED $O_3$ than in INTERACT $O_3$. In contrast, the temperature trends are weaker in FIXED $O_3$ than in INTERACT $O_3$. We therefore conclude that the difference in the imposed ozone trends cannot explain the difference in the temperature trends. 3) The CMIP6 ozone field is*

*interpolated from monthly values to the model time step and the studies of Sassi et al. (2005) and Neely et al. (2014) showed that this can lead to lower temperature trends when monthly ozone fields are prescribed. In this case, the SW heating rate trend would also be weaker in FIXED $O_3$ than in INTERACT $O_3$. However, the FIXED $O_3$ SW heating rate trend is stronger in our study, in line with the stronger ozone trend. Therefore, the monthly resolution of the prescribed CMIP6 ozone field cannot explain the weaker temperature trend in FIXED $O_3$. 4) Feedbacks between ozone, radiation and dynamics cannot occur in FIXED $O_3$. In a recent study using a different model, Haase et al. (2020) used the daily three-dimensional ozone from the interactive chemistry version and prescribed it to the version without interactive chemistry. Despite the fact that the ozone and SW heating rates were the same in their two ensembles, they still found differences between the SH polar cap lower stratospheric temperature and dynamical heating rate trends and attributed these differences to the missing ozone-related feedbacks when ozone is prescribed. These ozone-radiative-dynamical feedbacks are also missing in our FIXED $O_3$ ensemble and might therefore contribute to the differences in the stratospheric temperature and dynamics trends between FIXED $O_3$ and INTERACT $O_3$."* (lines 844-864)

A further, less fundamental question relates to the treatment of radiatively active gases other than ozone in CHEM-OFF. How do you treat water vapour, methane, and nitrous oxide when chemistry is turned off, in such a way as to minimize differences between the CHEM-OFF and CHEM-ON simulations? Or is chemistry running in all model variants and just a different flavour of ozone is fed into radiation?

*All other radiatively active gases are also prescribed in CHEM OFF (now termed FIXED $O_3$), except for water vapour. In the troposphere water vapor varies with cloud formation, among others. In the stratosphere and mesosphere the major source of water vapor, besides transport from the troposphere, is the oxidation of methane. In order not to underestimate middle atmospheric water vapor, ECHAM6 includes a submodel that parameterizes methane oxidation as well as the photolysis of water vapor in FIXED $O_3$. The ozone chemistry is the main difference between INTERACT $O_3$ and FIXED $O_3$ and it is therefore reasonable to expect that this is the driver of the differences in stratospheric dynamics between the two ensembles.*

And finally, you only mention $CO_2$ and methane as 'GHGs' in the set-up of the 'NoODS' simulations. How about $N_2O$? Is that considered an ODS? Technically it is, but it affects ozone very differently from the halogenated ODSs (see e.g. https://doi.org/10.5194/acp-18-1091-2018). Typically ODSs are taken to be chlorinated and brominated halocarbons controlled by the Montreal Protocol. Please clarify.

*Yes, we consider $N_2O$ to be an ODS. We adapted the text to state this clearly, now at line 188: "Here, we use GHG to refer to $CO_2$ and $CH_4$ only, while the other anthropogenic GHGs, including $N_2O$, fall under the ODS category."*

Final question: Are you planning to contribute this model to CMIP6, given the large effort put into producing the PI spin-up and the historical simulations?

*Unfortunately, we cannot contribute this model to CMIP6 as we do not have enough human resources, the storage capacity and funding to perform and store the minimum number of experiments required to be part of CMIP6.*

Minor comments:

P2L29: "at destroying ozone"

*We rephrased as suggested, thank you for pointing this out. (Now at line 35)*

P2L52: I think it's controversial whether East Antarctica actually experienced cooling, considering the difficulties with measuring temperature there (distinguishing cloud from ice in IR-measurements, sparsity of ground-based measurements). For a "greyliterature" comment on this see http://www.realclimate.org/index.php/archives/2004/12/antarctic-cooling-global-warming/. It's more robust to assert that the rate of warming in Antarctica exhibits large regional variations and that a large warming of much of West Antarctica has occurred.

*The statement now reads: "The formation of the ozone hole also affected the Antarctic surface temperatures, with large regional variations in the temperature trend over the continent. Significant warming over the Antarctic Peninsula and Patagonia was reported by Thompson and Solomon (2002)." (lines 57-60)*

P3L75: I find "zonally oriented asymmetries" confusing. I suggest dropping this phrase and just call them "zonal asymmetries".

Here, we wished to first define the term "zonal asymmetries". We rephrased to "asymmetries in the zonal direction" to make it more clear.

P4L127: Worth noting that these were not just "different climate models" but actually two different chemistry-climate models whose results for "historical" ozone were averaged to form the CMIP6 ozone forcing dataset. You want to cite https://doi.org/10.1002/2017GL076770 for this.

*We noted this at line 148 "Additionally, the prescribed ozone field, which was generated by averaging the output of two different CCMs…"*

P6L1: I infer from "CO$_2$ emissions" that this model is run using an interactive carbon cycle? Please state in the text if that is the case. Using methane "emissions" is also unusual in this context; typically methane VMRs are prescribed at the surface. "Emissions" are typically only used for tropospheric ozone and aerosol precursors. Also, please state explicitly how N$_2$O, HFCs, PFCs and other minor greenhouse gases are treated (see above).

*As the FOCI version used in this study does not include ocean biogeochemistry, the carbon cycle is not closed. Hence our wording is wrong. We indeed prescribe surface volume mixing ratio's of CH4, CO2, N2O and all minor greenhouse gases which the simplified chemical mechanism is able to handle. We updated the text, now found at line 183, and replaced "...in which emissions of both GHG and ODS..." with "...in which surface volume mixing ratios of both GHG and ODS are prescribed and vary as a function of time...".*

P10L298: "indicates"

*We corrected the spelling mistake, thank you for finding it.*

P13L407ff: I can see westward shifts in CHEM-ON and CHEM-OFF (figure 6), but the difference in the rates of progression is not obviously discernible to me. Could you perhaps think of a way of visualizing this better, and perhaps formalize that the trends are significantly different? You may want to cite https://doi.org/10.5194/acp-17-14075-2017 here.

*Fig. 6 now includes timeseries of the longitude at which the ozone maximum zonal anomaly occurs within the ridge region (panel f) and at which the ozone minimum occurs within the trough (panel g) for the two ensembles. This makes it easier to see where the difference in the rate of eastward progression of the ozone wave comes from. While both ensembles exhibit a similar eastward shift of the ridge (within their uncertainty bounds), only CHEM OFF (FIXED $O_3$) exhibits a significant eastward shift of the trough. The difference between eastward progression of the trough in the two ensemble is now clear, and it is statistically significant, as the 95% confidence intervals do not overlap. The fact that the CHEM ON (INTERACT $O_3$) trough does not shift with time can also be visualized in Fig. 8a, which shows the trend in the ozone anomalies from the zonal mean (note that, at the request of reviewer #3, we changed the period over which we compute the trends to 1958-2002). The trough of the trend exhibits similar magnitudes on both sides of the time mean ozone wave trough.*

*The additional panels in Fig. 6 are discussed in Sect. 4.1 at lines 451-457. We cited the results of Dennison et al. (2017) at line 462, as suggested.*

P16L507: "shifted"

*Thank you for pointing out this spelling mistake, we corrected it.*

Figure 4 and elsewhere: My inclination is to avoid introducing scaling factors into colour bars (e.g. $10^{-2}$ m s$^{-1}$, $10^{-4}$ m s$^{-1}$) and just state unscaled units (cm/s, mm/s). This is also done elsewhere but does not improve clarity, I find. Also figure 5, panel a. Why not have the scale range from -0:36 to +0:36?

*We removed the scaling factors from Figs. 4, 5, 6, 7 and 12 and changed the units accordingly.*

**Anonymous Referee #3**

The paper reports on the impact of ozone depletion and greenhouse gases on atmospheric circulation trends in the Southern Hemisphere in the new FOCI model. In addition, the paper also documents the impact of prescribing (rather than simulating) an ozone field on the effects of ozone depletion, by comparing ensembles with interactive ozone against ensembles using the CMIP6 ozone forcing. The authors conclude that FOCI captures the effects of ozone depletion and GHGs on the circulation. In addition, they also conclude that prescribing ozone rather than simulating one interactively leads to a weaker tropospheric response to ozone depletion. Based on these results, the paper claims that climate models prescribing CMIP6 ozone will underestimate the historical ozone-induced dynamical changes in the Southern Hemisphere. The subject of the paper is of relevance and interest for the readership at Atmospheric Chemistry and Physics. The paper is well written and the analysis is detailed and nicely discussed. However, some of the evidence provided is not convincing and as a result, some of the implications of this paper (models without interactive chemistry underestimating the effects of ozone depletion) are over-stated, especially concerning the limitations of imposing a prescribed ozone in historical simulations. The authors need to provide more convincing evidence to support some of the claims, or substantially revise some of them (perhaps tone them down!). Hence, I recommend major revisions, as detailed below.

MAJOR ISSUES:

1) This paper does not really provide a clean isolation of the 'ozone feedback'. Ozone from CMIP6 is substantially different from the ozone simulated by FOCI, as shown in Fig.7. This leads to a systematic bias in the 'CHEM OFF' experiments, as discussed in section 4.1. In addition to differences in climatological values, trends in the prescribed and interactive ozone are also different. Hence, any effects on the variability/trends are a result of these differences, rather than a missing 'ozone-radiative-chemical feedback' in the CHEM OFF ensemble (e.g., L16, L607). If the authors wish to quantify the ozone-chemical feedback, they should compare ensembles using the interactive ozone vs ensembles imposing a (time-varying) ozone, derived from the same model system (FOCI), rather than from other models (CMIP6).

*It is true that our set-up does not allow us to isolate the effect of missing feedbacks involving ozone, radiation and dynamics from effects arising from prescribing a different ozone field in a clear manner. We thank the reviewer for pointing this out. The main purpose of our study is to compare how the same model, FOCI, performs in simulating the effects of ozone depletion when it is run in its interactive chemistry configuration and when it is run with prescribed CMIP6 ozone. The aim of the study is not to isolate the various feedbacks involving ozone.*

*While our study cannot separate the effects of the ozone feedbacks from those due to the different ozone climatologies and trends, we also cannot completely exclude the possibility that the missing feedbacks play a role in setting the reported trend differences, in addition to the fact that the prescribed ozone climatology is different and the ozone hole is not consistent with the simulated dynamics. The importance of these feedbacks was recently shown in the study of Haase et al., (2020) using a different model (CESM-WACCM), in which three ensembles of nine simulations each are compared: one ensemble using interactive ozone chemistry (Chem ON), one ensemble in which the daily zonal mean ozone from Chem ON is prescribed (Chem OFF) and one ensemble in which the daily three dimensional ozone from Chem ON is prescribed (including zonal asymmetries in ozone, Chem OFF 3D). While the ozone and shortwave heating rate fields are the same in Chem ON and Chem OFF 3D, the authors still found differences in the lower stratospheric temperature and dynamical heating rate trends between the two ensembles (their figure 12) and attributed these differences to the fact that feedbacks involving ozone are not represented in Chem OFF 3D. Since these feedbacks are also missing from our simulations with prescribed CMIP6 ozone, the results of Haase et al. (2020) suggest that they also contribute to the different lower stratospheric trends that we find in our simulations with prescribed and interactive chemistry. We agree that they are not the sole and probably not the most important contributor and we revised the phrasing throughout the manuscript to put less emphasis on these feedbacks that we cannot prove with the setup of our FOCI simulations. We also included a new paragraph in Sect. 5 at lines 844-864 discussing all the possible reasons for the differences in the SH stratospheric response to ozone depletion in our FOCI ensembles.*

*Regarding the influence of the different ozone trends in CMIP6 and FOCI, Figs. 7, 10 and S4 bring strong evidence that the different ozone trends in the interactive and prescribed ozone ensembles cannot explain the differences in temperature trends. While the ozone and SW heating rate trends are stronger in FIXED $O_3$ (former Chem OFF), the temperature trend is weaker.*

*"Several factors can potentially explain the differences in the ozone-induced stratospheric temperature and circulation trends between INTERACT $O_3$ and FIXED $O_3$. 1) The CMIP6 and FOCI ozone fields exhibit different climatologies, as discussed in Sect. 4.1. Neither the climatological CMIP6 ozone hole, nor its variability are consistent with FOCI's dynamics. The fact that the prescribed ozone hole is displaced in relation to the simulated polar vortex alters the propagation of planetary waves from the troposphere to the stratosphere and therefore leads to*

*changes in the dynamical response to ozone depletion. This results in different dynamical heating rate trends in FIXED O₃ and INTERACT O₃. 2) The CMIP6 and FOCI ozone fields exhibit different trends. The austral spring polar cap ozone and, consequently, the SW heating rate trends are stronger in FIXED O₃ than in INTERACT O₃. In contrast, the temperature trends are weaker in FIXED O₃ than in INTERACT O₃. We therefore conclude that the difference in the imposed ozone trends cannot explain the difference in the temperature trends. 3) The CMIP6 ozone field is interpolated from monthly values to the model time step and the studies of Sassi et al. (2005) and Neely et al. (2014) showed that this can lead to lower temperature trends when monthly ozone fields are prescribed. In this case, the SW heating rate trend would also be weaker in FIXED O₃ than in INTERACT O₃. However, the FIXED O₃ SW heating rate trend is stronger in our study, in line with the stronger ozone trend. Therefore, the monthly resolution of the prescribed CMIP6 ozone field cannot explain the weaker temperature trend in FIXED O₃. 4) Feedbacks between ozone, radiation and dynamics cannot occur in FIXED O₃. In a recent study with a different model, Haase et al. (2020) used the daily three-dimensional ozone from the interactive chemistry version and prescribed it to the version without interactive chemistry. Despite the fact that the ozone and SW heating rates were the same in their two ensembles, they still found differences between the SH polar cap lower stratospheric temperature and dynamical heating rate trends and attributed these differences to the missing ozone-related feedbacks when ozone is prescribed. These ozone-radiative-dynamical feedbacks are also missing in our FIXED O₃ ensemble and might therefore contribute to the differences in the stratospheric temperature and dynamics trends between FIXED O₃ and INTERACT O₃." (lines 844-864)*

2) The comparison with observations is missing. The paper claims in several instances that the FOCI model 'reliably' captures the effects of ozone depletion (e.g., in the Abstract on L17) and that in simulations with interactive chemistry, the effects of ozone depletion are stronger and closer to the observations (e.g. in the Conclusions near L610). However, this comparison with observations is lacking, as no single observational data-set is shown in the paper, using the same analysis period & statistical method. The authors should directly compare their ensembles (especially REF and CHEM ON) against observations, at least for some of the large-scale circulation metrics, such as zonal mean zonal wind, temperature, near-surface wind (850 hPa), to build confidence in some of their claims regarding the model's skills in capturing observed trends and regarding the increased 'realism' in the simulations with interactive chemistry. I suggest using ERA5 or some other high-quality re-analysis product for this purpose.

*We updated the manuscript to include the temperature trends obtained from the IGRA radiosonde data set (Figs. 10c and S5d) and from the ERA5 reanalysis (Figs. 9c, 10c, S5h), as well as the ERA5 zonal wind temperature trends (Figs. 9f, 10d-f, 11e, f).*

*Both ERA5 and IGRA agree well with the FOCI simulations, showing the well-known pattern of ozone-induced cooling in the Antarctic lower stratosphere between October and December, which peaks in November around 100 hPa (Figs. 9 and S5). The magnitude of the ERA5 trend is a bit larger than that of the IGRA trend, but both fall at the lower end of the INTERACT O₃ trend range and both exceed the FIXED O₃ trends (Fig10c). We discuss these results in Sect. 4.2.1 (lines 546-561) and therefore conclude at lines 564-566: "…FIXED O₃ tends to underestimate the ozone-induced cooling, while INTERACT O₃ tends to overestimate it, although individual ensemble members can simulate trends that are close to those observed."*

*The polar vortex trends in FIXED O₃ are closer to the ERA5 trends than the INTERACT O₃ trends are (Fig. 10d). The FIXED O₃ ensemble mean trends in the tropospheric jet`s strength and position also agree better with ERA5 in the zonal mean, however the ERA5 trend in the jet`s*

*strength is not statistically significant in the zonal mean (Fig. 10 e and f). Please see Sect 4.2.2 for a detailed discussion of the trends in the westerly jets.*

*Our conclusion was edited accordingly:*

*"– The ozone-induced austral spring polar cap cooling in the lower stratosphere is weaker in FIXED O₃ than in INTERACT O₃. The cooling trends estimated from the IGRA radiosonde observations and from the ERA5 reanalysis are stronger than those obtained from the FIXED O₃ simulations and fall at the lower end of the range of trends simulated by INTERACT O₃.*
*– The acceleration of the stratospheric jet in response to ozone depletion is also weaker in FIXED O₃ than in INTERACT O₃ and it agrees better with the estimate from the ERA5 reanalysis. In contrast, the tropospheric jet trend differences between FIXED O₃ and INTERACT O₃ fall within the range of internal variability." (lines 776-782)*

3) Inappropriate time-period chosen to analyze the impact of ozone depletion. The authors use the 1958-2013 period to calculate trends in their historical simulations, and they derive the impact of the ODS by taking differences between noODS and REF ensembles (REF is presumably the same as the CHEM ON...??). ODS emissions were phased out in the mid 90s and as a result, ozone depletion trends stopped near the year 2000. Since the beginning of the 21st century, we have already seen the emergence of 'healing' in the ozone layer (Solomon et al., 2016). Recently, it has also been shown that this resulted in a change in the tropospheric circulation trends in the SH (Banerjee et al., 2020). Hence, the trends calculated in this paper do not properly isolate the effects of ozone depletion, as the trends before and after 2000 are probably very different. I would strongly recommend choosing an earlier end-date in the analysis of trends (e.g. early 2000s).

*We carefully compared the trends for 1958-2013 to trends for periods starting in 1958 and ending around the year 2000. The magnitude of the trends increased for earlier end years and we therefore settled on the period 1958-2002, a period over which ozone depletion is strong and that is long enough to allow the clear detection of trends from the internal variability in the model. As austral spring ozone levels above Antarctica remain low in 2013 compared to the pre-ozone depletion levels, despite the small signs of ozone recovery, the features of the trends over the new period remained similar. Ozone depletion is the dominating driver of change in the Southern Hemisphere over the past decades, detectable irrespective of the end year chosen.*

*For the difference between REF and NoODS or NoGHG, we settled on the period 1978-2002, characterized by the strongest ozone depletion. This period was chosen in order to balance the need for a long-enough period to be able to isolate statistically significant effects of ozone depletion from the strong internal variability in the model, with the recommendation to choose an end date around the year 2000. Unlike the case of INTERACT O₃ and FIXED O₃, where linear trends are compared, we compare differences in daily or monthly climatologies of various fields in REF and in NoODS/NoGHG in order to isolate the effects of ozone depletion and increase in GHG. Therefore, the period 2002-2013 would have been important, as the temperature and zonal wind climatologies, as well as the climatologies of other fields affected by ozone depletion, are very different in REF and NoODS during this period. This is because, as mentioned above, although ODS levels started to decrease and there are signs of ozone recovery, the state of the Antarctic ozone in 2013 as well as that of the stratospheric dynamics affected by the ozone hole are far from resembling their state prior to ozone depletion. On the other hand, the period before 1978, when the trends are weak, is characterized by similar climatologies for the three ensembles. Therefore we considered the period 1978-2002 to be a good compromise. As shown by the new Figs. 1-4, the results of Sect. 3 have remained similar to those for the 1958-2013 period.*

*REF and INTERACT O₃ (former CHEM ON) are distinct ensembles, the former also including a high-resolution ocean nest, as explained at lines 194-196 and in the response to the comment regarding L160-180.*

4) No convincing statistics. Aside from direct comparison with observations, we need to make sure differences are really robust across ensemble members. Several studies (e.g., Seviour et al., 2017) have shown how large the variability in the SH can be, and how it can explain differences across transient simulations. Can we make sure the CHEM ON vs OFF differences are really robust in light of this large variability? I would strongly recommend comparing the response to ozone depletion (and most importantly, the CHEM ON vs OFF differences) against the inter-ensemble spread. Ideally, the authors should show the individual members of all ensembles against observations, to give more confidence in the two key statements made by this paper concerning (1) this model reliably reproducing observations in terms of the effects of ozone depletion and (2) its trends being significantly weaker with prescribed (CHEM OFF) than interactive (CHEM ON) ozone.

*We added two new figures in the manuscript (Figs. 10 and 13) and three new figures in the supplement (Figs. S3, S5 and S6) showing the spread of the INTERACT O3 and FIXED O3 ensemble members regarding the ozone-induced SH trends in temperature and dynamics.*

*In the stratosphere, there is a clear distinction between the range of temperature (Figs. 10c, S3 and S5) and zonal wind (Figs. 10d, S6) trends simulated by the INTERACT O₃ members and those simulated by the FIXED O₃ members, with all members of the latter ensemble simulating weaker trends than all members of the former ensemble. This also holds true for the trends in the eddy heat flux, EP flux divergence, V\*, W\* and dynamical heating rate in the middle and upper stratosphere shown in Fig. 13. Therefore, the INTERACT O₃ ozone-induced austral spring stratospheric trends in temperature and dynamics are significantly stronger than the FIXED O₃ trends.*

*The situation is indeed less clear in the troposphere, where it seems that there is also a different timing when the zonal wind trend maximizes and where we cannot clearly distinguish the difference between the INTERACT O₃ and FIXED O₃ westerlies trends from the strong internal variability. We make this clear in the paragraph at lines 639-646.*

*Please also see our response to major comment 2 regarding the comparison to observations and reanalysis.*

5) Lacking discussion of relevant literature. The authors do not sufficiently discuss some key studies, which already looked at role of interactive chemistry in simulating the impact of ozone depletion. One of them was Eyring et al., 2013, which directly compared CMIP5 simulations from CHEM and NOCHEM models. More recently, Seviour et al. (2017) and S. Woo-Son et al. (2018) also extensively analyzed multi-model comparisons (CCMI and CMIP5) in terms of their simulated ozone depletion, and found barely any robust difference between models with and without interactive ozone. These papers should be properly cited and discussed, to provide a more critical and balanced discussion throughout the paper.

*Thank you for the suggesting these references. We included them in the introduction, as well as in the discussion on the trends in the tropospheric westerly jet. We note, however, that using different models to evaluate differences in the response to ozone depletion related to the method of imposing ozone makes it hard to assess how other differences between those models, such as the strength of the stratosphere-troposphere coupling, might influence the results. Furthermore,*

*in the study of Eyring et al. (2013), some of the models "with chemistry" actually used prescribed ozone, but the ozone was produced by the interactive chemistry version of the same model, rather than by other models, as it was the case in the group "without chemistry".*

*"In contrast, the tropospheric jet's response to ozone depletion is not significantly different between models with and without ozone chemistry in studies that used different models to assess the sensitivity of the response to how the ozone is imposed (Eyring et al., 2013; Seviour et al., 2017; Son et al., 2018)." (lines 135-137)*

*"Therefore, we conclude that the differences in the tropospheric westerly jet trends in INTERACT $O_3$ and FIXED $O_3$ are within the range of internal variability, in agreement with the results of Eyring et al. (2013), Seviour et al. (2017) and Son et al. (2018)." (lines 644-646)*

6) Unjustified claims regarding underestimation of the effects of ozone depletion in models without interactive chemistry (e.g. see L659). Several papers have shown that actually, imposing or simulating the ozone hole does not make a lot of difference. See for example S.-Woo-Son et al., 2018, as well as Eyring et al., 2013. While it's true that there inter-comparison studies do not cleanly isolate the impact of interactive chemistry alone, they do not see any systematic difference between both class of models (CHEM or NOCHEM), and they span over a wider range of uncertainty, since they look at many different models rather than a single model, as done in this paper. Hence, the implications of this study may be smaller than stated in the paper (e.g. the statement in L21-23 in the Abstract). Moreover, this paper does not properly compare any of the FOCI trends with observations (major comment 2), nor cleanly isolates the importance of interactive chemistry (major comment 1). Hence, the claims about CMIP6 models underestimating the historical ozone-induced changes in the SH are unjustified.

*These studies (Eyring et al., 2013, Seviour et al., 2017 and Son et al., 2018) focused mostly on the circulation in the troposphere when differentiating between models with and without interactive chemistry. Changes in the stratospheric residual circulation, for example, are not discussed in these studies. Our (extended) results for the tropospheric westerly jet agree with the results of the cited studies and we acknowledge this in the revised version of the manuscript at lines 644-646 (reproduced in the response to major comment 5). For the stratosphere, our results show that simulations with prescribed CMIP6 ozone exhibit weaker trends in both temperature and dynamics than simulations with interactive ozone chemistry (Sect. 4.2). We included temperature and wind trends from the IGRA radiosonde observations and the ERA5 reanalysis. The observed temperature trends agree better with those in INTERACT $O_3$, while the observed wind trends agree better with those in FIXED $O_3$. We made changes throughout the manuscript in line with the new results. Please also see our response to major comments 1, 2 and 5.*

SPECIFIC ISSUES:

L16 "missing ozone radiative dynamical feedbacks" - see major comment 1: The CHEM ON vs OFF comparison rather quantifies the impact of a systematic bias, rather than a true feedback (which could only be quantified by comparing another CHEM OFF ensemble driven with the ozone forcing from CHEM ON).

*We removed the part of the sentence regarding the ozone-related feedbacks. Please also see our response to major comment 1.*

L21-23 In light of the discussion given above (points 1, 5-6), I frankly do not find this statement very convincing.

*The statement now reads "The results obtained with the FOCI model suggest that models which prescribe the CMIP6 ozone field still simulate a weaker Southern Hemisphere stratospheric response to ozone depletion compared to models that calculate the ozone chemistry interactively", in line with the results presented in Sect. 4.2 for the FOCI climate model.*

L65 Oehrlein et al. (2020) is another recent relevant paper that studied this problem, as they compare CHEM ON vs OFF experiments strictly having the same ozone climatology. I suggest adding this paper to the reference list.

*Thank you for pointing out this new publication. We included this reference, as suggested below, in the discussion of the effect of interactive chemistry on the frequency of SSWs (lines 92-94) and in the discussion of ozone related feedbacks (lines 118-119). However, as this study used time-slice simulations with constant year 2000 forcing and with prescribed climatological ozone, we cannot cite it when discussing the response to ozone depletion in models with interactive or prescribed ozone, which is the topic of lines 70-72 (former line 65): "Multiple lines of evidence suggest that the method used to specify stratospheric ozone in models affects their response to ozone depletion…"*

L83-85 Oehrlein et al., 2020 also explored this. They show that in time-slice simulations with constant forcing, the effect of interactive chemistry on SSWs frequency is not statistically significant. Adding this paper could help making the point about the lack of robustness across different studies on this.

*We added the citation at lines 92-94: "In a recent study, Oehrlein et al. (2020) found no significant difference in the number of midwinter SSWs between their 200-year time-slice simulations with interactive and with prescribed zonally symmetric ozone."*

L93 there were also papers showing the contrary, i.e. that models with/without interactive chemistry were very similar in their simulated trends. One paper showing this was Seok-Woo Son et al., 2018. This paper should be cited and discussed.

*Thank you for suggesting this paper. We add the reference in the introduction (lines 135-137) and in the discussion of the trends in the tropospheric jet (lines 644-646). Please also see our response to major comment 5. We also note that this study actually found different polar vortex trends between models with and without interactive chemistry (their Fig. 3), in agreement with our results, and only the trends in the tropospheric circulation were similar.*

L114-118 Oehrlein et al., 2020 also studied this problem, using time-slice rather than transient simulations, partly confirming some of the results of Haase and Matthes (2019) but also refuting some others (e.g. the influence on SSW frequency), so I recommend citing this paper here, too.

*We added the citation at line 118-119 "The importance of such feedbacks in both hemispheres was previously shown in the studies by Haase and Matthes (2019), Haase et al. (2020) and Oehrlein et al. (2020)."*

L125-140 Another problem which is not discussed at all is the vertical interpolation. Interpolating the ozone forcing from CMIP6 which is provided on pressure levels on FOCI's own z-levels may create errors, which can be non-negligible near the tropopause. This can create problems with

radiative transfer, as e.g. discussed in Hardimann et al., 2019. Have the authors tested whether this happens, too?

*Thank you for pointing us to this interesting paper. No, we have not tested whether a possible mismatch between the prescribed ozone and the simulated tropopause temperatures may introduce unphysical radiative heating or cooling around the tropopause. As Hardiman et al., (2019) state, the largest mismatch occurs when performing +4K simulations and not when doing historical or scenario simulations, as the ozone forcing dataset accounts for changes in tropopause height. Hence, we do not expect that the smaller mismatch between the tropopause height in FOCI and the one present in the ozone forcing dataset is the key difference between CHEM ON (now termed INTERACT $O_3$) and CHEM OFF (now termed FIXED $O_3$). We added a sentence at lines 150-152 to make the reader aware that the problem described in Hardiman et al., (2019) can partially explain the difference between the two ensembles: "Moreover, Hardiman et al. (2019) showed that a mismatch between the tropopause height present in the prescribed ozone dataset and the tropopause height in the climate model that uses the prescribed ozone dataset can cause erroneous heating rates around the tropopause."*

L139 "correctly simulate the effects of ozone depletion" - the authors do not show any observations in this paper. Hence, we cannot really determine whether the REF ensemble (which is the same as CHEM ON I guess?) is really close to observations and whether CHEM OFF is systematically off. I strongly suggest adding one such analysis. This could be, e.g. by adding trends in jet-latitude or SAM trends and show individual ensembles vs observations, as done e.g. in Seviour et al., 2017.

*We added the temperature trends obtained from the IGRA radiosonde data set (Figs. 10c and S5d) and from the ERA5 reanalysis (Figs. 9c, 10c, S5h), as well as the ERA5 zonal wind and temperature trends (Figs. 9f, 10d-f, 11e, f). The difference between the REF and NoODS ensembles, which gives the effect of ozone depletion, captures the changes in lower stratospheric temperature, in the polar vortex and in the tropospheric westerly jet shown in observations. The magnitude of the changes cannot be directly compared to observations, however, as the latter also include the effects of global warming. Therefore, we removed the word "correctly" from the statement. The trends in the INTERACT $O_3$ and FIXED $O_3$ ensembles are compared to observations in Sect. 4.2. Please also see our response to major comment 2 regarding the comparison of the trends simulated by the FOCI ensembles to those obtained from IGRA and ERA5.*

*The INTERACT $O_3$ (former Chem ON) and REF ensembles are different, please see our response to the comment below.*

L160-180 if FOCI by default uses interactive chemistry in the REF experiment, then what is the difference between this ensemble and the CHEM ON ensemble?

*All the ensembles discussed in Sect. 3 (REF, NoODS, NoGHG) include interactive chemistry. The difference between REF and CHEM ON (now termed INTERACT $O_3$) is that REF additionally includes a high-resolution ocean nest around South Africa (line 195). The nest was included to enable the study of changes in the Indo-Atlantic water exchange, known as the Agulhas Leakage, which takes the form of mesoscale eddies, rings and filaments. This is, however, beyond the scope of this study.*

*We added the information that REF and INTERACT $O_3$ differ at line 195-196: "Therefore, the REF ensemble differs from the INTERACT $O_3$ ensemble discussed below."*

L210-220 Is these multiple filtering really needed? Are the results for the SAM sensitive to the way the data is filtered? It would be nice if the authors could comment on this.

*We calculated the SAM using the well-established method of Gerber et al. (2010). The filtering serves the purpose of removing a slowly varying seasonal cycle from the geopotential height field, such that the resulting SAM index reflects internal variability and is suitable for analyzing timescales. Gerber et al. (2010) provided a comparison of the annular modes obtained with their method and the annular modes obtained using the method of Baldwin and Thompson (2009), who used a fixed seasonal cycle to define the geopotential height anomalies. The method of Gerber et al. (2010) for computing the annular modes was used in previous studies investigating their timescales (Simpson et al., 2011, Dennison et al., 2015, Simpson and Polvani 2016, Haase et al., 2020) as well as other studies that examined other aspects of the annular modes (Charlton-Perez et al., 2013, Haase and Matthes, 2019, Simpson et al., 2020).*

*Charlton-Perez, A. J., et al. (2013), On the lack of stratospheric dynamical variability in low-top versions of the CMIP5 models, J. Geophys. Res. Atmos., 118, 2494– 2505, doi:10.1002/jgrd.50125.*

*Isla R. Simpson, Lorenzo M. Polvani (2016), Revisiting the relationship between jet position, forced response, and annular mode variability in the southern midlatitudes, Geophysical Research Letters, 10.1002/2016GL067989, **43**, 6, (2896-2903).*

*Isla R. Simpson, Julio Bacmeister, Richard B. Neale, Cecile Hannay, Andrew Gettelman, Rolando R. Garcia, Peter H. Lauritzen, Daniel R. Marsh, Michael J. Mills, Brian Medeiros, Jadwiga H. Richter (2020), An Evaluation of the Large-Scale Atmospheric Circulation and Its Variability in CESM2 and Other CMIP Models, Journal of Geophysical Research: Atmospheres, 10.1029/2020JD032835, **125**, 13.*

L231 see major comment (3) concerning the time-period. The authors should explore the sensitivity of these results to the end year chosen, and a shorter period (e.g., 1958-2000) would probably be more appropriate to explore the effects of ozone depletion.

*We changed the period over which we compute the INTERACT O$_3$ (former Chem ON) and FIXED O$_3$ (former Chem OFF) trends to 1958-2002 and the period over which we compute the differences between REF and NoODS/NoGHG to 1978-2002. While the magnitude of the trends and of the differences changed, the conclusions of our study were not dependent on the end year chosen. Please also see our response to major comment 3.*

L256 Actually, this problem has been studies in multiple papers, which looked at the upper stratospheric ozone response to large CO2 forcings in detail; e.g. Haigh and Pyle (1982), Jonsson et al. (2004), Chiodo et al. (2018). I suggest adding these papers here.

*Thank you for pointing us to these studies. They are now cited at lines 289-290.*

L262 This effect (GHGs –> polar cap ozone increase due to faster BDC) has been widely studied in the context of climate sensitivity experiments, imposing large CO2 forcings (2x and 4xCO2), such as e.g. Dietmuller et al. (2014); Nowack et al. (2015); Chiodo et al. (2018). Such results also apply here, although the GHG forcing studied in this paper is much smaller.

*We included these references at lines 296-298, thank you for pointing them out.*

L287 "realistically capture" - I recommend adding one figure for the observations, so that the reader can appreciate how close REF (or CHEM ON) are to the observational trend. The validation paper for FOCI (Matthes et al., 2020) did not really show trends in the SH circulation, so this should be done in this paper, since the main message is that FOCI is "able to capture" the impacts of ozone depletion.

*We added the SH westerlies trends from ERA5 in Figs. 9 and 11, similarly to Figs. 2 and 3. These trends are discussed in Sect. 4.2. FOCI simulates the SH effects of ozone depletion reported by previous studies (cited in the introduction as well as in Sect. 3) and seen in observations and reanalysis (Sect.4.2). We removed the word "realistically", as we cannot compare the magnitude of the ozone effects on the different fields to observations, since the observations also include the effects of increasing GHGs. Instead, we compared the total temperature and wind changes in the INTERACT $O_3$ and FIXED $O_3$ ensembles to observations in Sect. 4.2.*

L366 "adequately simulates" - same comment as on L287

*The residual circulation changes in response to ozone depletion simulated by FOCI were also reported by the previous studies cited in this paragraph. Given the agreement between all these studies, we find the phrasing, now found at lines 402-403, appropriate.*

L375 I would strongly recommend changing the labeling (CHEM OFF) to something more descriptive of what is really used here (CMIP6 O3 forcing). How about CHEM ON vs FIXO3 CMIP6?

*We changed the names of the ensembles from CHEM OFF to FIXED $O_3$ and from CHEM ON to INTERACT $O_3$. As we make it clear in Sections 1 and 2, as well as in the title, that the prescribed ozone comes from CMIP6, we preferred FIXED $O_3$ to FIXO$_3$ CMIP6 for brevity.*

L474 "agree better with observed trends" –> this has not really been shown here, so I am not convinced about this statement. To lend confidence on the results and statements like this throughout the paper, the authors would need to show (1) that all three ensemble members are closer to the observations than CHEM OFF and (2) that they are all significantly different from any of the members in CHEM OFF. This also applies to L610 in the conclusions section.

*We added the temperature trends from the ERA5 reanalysis and the IGRA radiosonde observations (Figs. 9, 10, S5), as well as the spread of the INTERACT $O_3$ and FIXED $O_3$ trends (Figs. 10, S3, S5). As seen in Fig. 10c, the IGRA and ERA5 temperature trends fall within the INTERACT $O_3$ range of trends, albeit in the lower end. All INTERACT $O_3$ temperature trends and both ERA5 and IGRA trends are stronger than all of the FIXED $O_3$ trends. The new figures are discussed in detail in Sect. 4.2.1 and in the reply to major comments 2 and 4. We deleted the statement at former line 474 and, based on the results shown in the new figures, conclude instead at lines 565-566 that "…FIXED $O_3$ tends to underestimate the ozone-induced cooling, while INTERACT $O_3$ tends to overestimate it, although individual ensemble members can simulate trends that are close to those observed."*

*The statement at former line 610 was deleted and instead we stated "The ozone-induced austral spring polar cap cooling in the lower stratosphere is weaker in FIXED $O_3$ than in INTERACT $O_3$. The cooling trends estimated from the IGRA radiosonde observations and from the ERA5 reanalysis are stronger than those obtained from the FIXED $O_3$ simulations and fall at the lower end of the range of trends simulated by INTERACT $O_3$." at lines 776-779 in the conclusion section.*

L494 "the feedbacks between ozone..." As stated in my major comment (1), this simulations set-up does not really cleanly isolate the feedback, as the CMIP6 ozone forcing leads to a systematically different basic state. How can we be sure that these differences are rather due to "biases" introduced by the CMIP6 ozone forcing, rather than a true "feedback"?

*The statement at former line 494 was removed. Instead, we discuss all possible reasons for the discrepancy in the INTERACT $O_3$ and FIXED $O_3$ trends in a new paragraph in the conclusion section at lines 844-864.*

*Please also see our response to major comment 1.*

L511 this is a very far fetched statement, given that some studies in the past have already shown that CMIP5 models with interactive chemistry (CHEM) do not significantly differ from those that impose the historical ozone forcing in terms of the tropospheric trends (e.g. Eyring et al., 2013; Seviour et al., 2017; Seak-Woo-Son et al., 2018... just to name a few!). To show that CMIP6 is different in this sense, the authors would need to use a different set-up and/or use more models. Otherwise, this is an over-statement which is not justified by the evidence provided in this paper. This also applies to L660-662.

*The statement at former line 511 was deleted. After we expanded our analysis as you suggested to show the trends in the individual ensemble members, we also conclude that only the trends in the stratospheric westerly jet are different between INTERACT $O_3$ and FIXED $O_3$, while the differences between the tropospheric jet trends fall within the range of internal variability, in agreement with the studies you cite.*

*"The tropospheric westerly jet trends in INTERACT $O_3$ and FIXED $O_3$ cannot be clearly differentiated, as it is the case for the stratospheric jet." (lines 639-640)*

*"Therefore, we conclude that the differences in the tropospheric westerly jet trends in INTERACT $O_3$ and FIXED $O_3$ are within the range of internal variability, in agreement with the results of Eyring et al. (2013), Seviour et al. (2017) and Son et al. (2018)." (lines 644-646)*

*"The acceleration of the stratospheric jet in response to ozone depletion is also weaker in FIXED $O_3$ than in INTERACT $O_3$ and it agrees better with the estimate from the ERA5 reanalysis. In contrast, the tropospheric jet trend differences between FIXED $O_3$ and INTERACT $O_3$ fall within the range of internal variability." (lines 780-783)*

*The whole paragraph ending at former line 662 was re-written (lines 835-843).*

L552-556 see major comment on L511, and major comment (1). This also applies to the statements on L606-607.

*We rephrased the statement at the former lines 552-556 to exclude the mention of the ozone-related feedbacks: "These different residual circulation changes are the consequence of the fact that the prescribed ozone hole is not consistent with the simulated dynamics." Please also see our response to major comment 1 on this issue. We now specifically refer to the stratospheric temperature and westerly winds trends, which were shown to differ between INTERACT $O_3$ and FIXED $O_3$ in Sect. 4.2 (see also our response to major comment 4).*

*The statement at former lines 606-607 was removed and replaced with "The acceleration of the stratospheric jet in response to ozone depletion is also weaker in FIXED $O_3$ than in INTERACT $O_3$ and it agrees better with the estimate from the ERA5 reanalysis. In contrast, the tropospheric jet trend differences between FIXED $O_3$ and INTERACT $O_3$ fall within the range of internal variability." (lines 780-783) The possible reasons for the stratospheric trend differences are discussed in a new paragraph at lines 844-864.*

REFERENCES

Banerjee et al., 2020; A pause in Southern Hemisphere circulation trends due to the Montreal Protocol, https://doi.org/10.1038/s41586-020-2120-4

Chiodo et al., 2018; The response of the ozone layer to quadrupled CO2 concentrations, DOI:10.1175/JCLI-D-17-0492.1

Dietmuller et al., 2014; Interactive ozone induces a negative feedback in CO2 driven climate change simulations, DOI:10.1002/2013JD020575

Haigh and Pyle, 1982; Ozone perturbation in a two dimensional circulation model, DOI: 10.1002/qj.49710845705

Hardimann et al., 2019; The Impact of Prescribed Ozone in Climate Projections Run With HadGEM3 GC3.1, DOI:10.1029/2019MS001714

Jonsson et al., 2004; Doubled CO2 induced cooling in the middle atmosphere: Photochemical analysis of the ozone radiative feedback, DOI:10.1029/2004JD005093

Nowack et al., 2015; A large ozone-circulation feedback and its implications for global warming assessments, DOI:10.1038/nclimate2451

Oehrlein et al., 2020; The effect of interactive ozone chemistry on weak and strong stratospheric polar vortex events, DOI:10.5194/acp-20-10531-2020

Seok-Woo Son et al., 2018; Tropospheric jet response to Antarctic ozone depletion: An update with Chemistry-Climate Model Initiative (CCMI) models, DOI:10.1088/1748-9326/aabf21

Seviour et al., 2017; Robustness of the Simulated Tropospheric Response to Ozone Depletion, DOI:10.1175/JCLI-D-16-0817.1

Solomon et al., 2016; Emergence of healing in the Antarctic ozone layer, DOI: 10.1126/science.aae0061